# Post-covid medical complaints following infection with SARS-CoV-2 Omicron vs Delta variants

Karin Magnusson [1,2] ✉, Doris Tove Kristoffersen[1], Andrea Dell'Isola[2], Ali Kiadaliri[2,3], Aleksandra Turkiewicz [2], Jos Runhaar[4], Sita Bierma-Zeinstra[4,5], Martin Englund [2], Per Minor Magnus[1] & Jonas Minet Kinge[1,6]

The SARS-CoV-2 Omicron (B.1.1.529) variant has been associated with less severe acute disease, however, concerns remain as to whether long-term complaints persist to a similar extent as for earlier variants. Studying 1 323 145 persons aged 18-70 years living in Norway with and without SARS-CoV-2 infection in a prospective cohort study, we found that individuals infected with Omicron had a similar risk of post-covid complaints (fatigue, cough, heart palpitations, shortness of breath and anxiety/depression) as individuals infected with Delta (B.1.617.2), from 14 to up to 126 days after testing positive, both in the acute (14 to 29 days), sub-acute (30 to 89 days) and chronic post-covid (≥90 days) phases. However, at ≥90 days after testing positive, individuals infected with Omicron had a lower risk of having any complaint (43 (95% CI = 14 to 72) fewer per 10,000), as well as a lower risk of musculoskeletal pain (23 (95%CI = 2-43) fewer per 10,000) than individuals infected with Delta. Our findings suggest that the acute and sub-acute burden of post-covid complaints on health services is similar for Omicron and Delta. The chronic burden may be lower for Omicron vs Delta when considering musculoskeletal pain, but not when considering other typical post-covid complaints.

An increase in medical complaints following mild SARS-CoV-2 infection, sometimes referred to as "long-covid", has been reported[1–3]. Although the SARS-CoV-2 Omicron variant has been associated with less severe acute disease and a reduced risk of hospitalizations compared with Delta, concerns remain as to whether long-term complaints persist to a similar extent as for earlier variants[4,5].

Given the increased secondary attack rate when the index case has Omicron rather than Delta[6], and the expectancy of many symptomatic but less serious cases even among vaccinated individuals[7], there is a need for knowledge of post-Omicron risks for medical doctors, health personnel and health policy makers. If the Omicron variant leads to temporary or persistent

post-covid complaints, it may impose a large burden on healthcare and society.

Survey data have been used to determine patterns of symptom persistence following SARS-CoV-2 infection, however, the estimates vary extensively, and they cannot be used to infer on the consequences for the healthcare services. For example, dyspnoea after recovery from primary SARS-CoV-2 infection has been reported in 10–20% of patients in one survey study and up to 75% of patients in another survey study[8,9]. Reporting and response biases will affect the accuracy of both symptoms and testing, leading to questionable validity and difficulties with comparisons between studies. Nordic National register data is based on healthcare services that are freely available for all inhabitants,

[1]Norwegian Institute of Public Health, Oslo, Norway. [2]Clinical Epidemiology Unit, Orthopedics, Department of Clinical Sciences Lund, Lund University, Lund, Sweden. [3]Centre for Economic Demography, Lund University, Lund, Sweden. [4]Department of General Practice, Erasmus MC University Medical Center Rotterdam, Rotterdam, The Netherlands. [5]Department of Orthopedics & Sports Medicine, Erasmus MC University Medical Center Rotterdam, Rotterdam, The Netherlands. [6]Department of Health Management and Health Economics, University of Oslo, Oslo, Norway. ✉e-mail: karin.magnusson@fhi.no

i.e. a medical record as seen in primary care represents both an indication of a complaint as experienced by the patient (and judged so by the medical doctor), and it represents a healthcare contact placing a certain demand on the healthcare services.

The linkage of such medical record data to data on variant-specific SARS-CoV-2 infection including the recently emerged Omicron variant can provide insights into both post-covid etiology and the expected burden on healthcare systems when many are vaccinated and have mild disease courses. Thus, we have studied whether individuals infected with the Omicron variant have an altered risk of post-covid complaints compared with (1) individuals infected with Delta, and (2) individuals who are non-infected. We also provide estimates of prevalent complaints for the acute, sub-acute and chronic post-covid phases, including data beyond 3 months after positive test.

## Results

Of in total 3,696,005 persons eligible for the study, 105,297 persons tested negative during our study period, and 57,727 persons had a positive test result that was screened for SARS-CoV-2 variant (Fig. 1). Individuals infected with the Omicron variant ($N = 13,365$) were generally younger, had higher education, fewer comorbidities and were more often vaccinated than individuals infected with the Delta variant ($N = 23,767$) (Table 1). There were also some group differences in the amount of follow-up time by study group and by outcome, in the main analysis and in the sensitivity analysis (S-Table 2). Among individuals testing negative and individuals who were untested, 18,866 (17.9%) and 121,317 (10.3%) tested positive during follow-up and were non-censored in the main analyses (test negative) and in the sensitivity

### Table 1 | Descriptive characteristics

| | SARS-CoV-2 Omicron variant | SARS-CoV-2 Delta variant | Testing negative for SARS-CoV-2 |
|---|---|---|---|
| | 13,365 | 23,767 | $N = 105,297$ |
| Age, median (interquartile range) | 36 (26–48) | 40 (32–49) | 42 (30–53) |
| Women, n (%) | 6638 (49.7) | 11,711 (49.3) | 52,905 (50.2) |
| Primary school[a], n (%) | 2886 (21.7) | 5183 (21.9) | 21,341 (20.3) |
| Upper secondary school[a], n (%) | 4179 (31.4) | 7821 (33.0) | 35,848 (34.2) |
| College/university[a], n (%) | 5461 (41.0) | 8948 (37.8) | 41,180 (39.3) |
| Born in Norway, n (%) | 9520 (71.2) | 16,138 (67.9) | 79,686 (75.7) |
| ≥2 comorbidities[b], n (%) | 1169 (8.7) | 2604 (11.0) | 12,858 (12.2) |
| One vaccine dose[c], n (%) | 12,463 (93.2) | 18,822 (79.2) | 97,964 (93.0) |
| Two vaccine doses[c], n (%) | 12,119 (90.7) | 18,011 (75.8) | 94,183 (89.4) |
| Three vaccine doses[c], n (%) | 1421 (10.6) | 1430 (6.0) | 17,362 (16.5) |
| Nr. of negative tests[d], median [IQR] | 2 [1–4] | 2 [1–4] | 1 [0–3] |
| Nr. of all-cause previous visits to primary care[e], median [IQR] | 2 [1–3] | 2 [1–4] | 2 [1–3] |

All characteristics measured prior to test.
[a]Highest achieved education level by end of 2019.
[b]Based on counts of medical records in specialist care, January 1, 2020 to December 7, 2021.
[c]The current vaccination status against COVID-19 before selected test date, based on records in vaccination database.
[d]Counts of registered tests, from January 1, 2020 to December 7, 2021.
[e]Counts of contacts with general practitioner or emergency ward, from January 1, 2020 to December 7, 2021.

analyses (untested), respectively. When these two study groups were combined, 140,183 (10.9%) tested positive and were censored in analyses with censoring of observations from the date of positive test and onwards. The mortality during follow-up was low (0.07% (95% CI = 0.03–0.13), 0.05% (95% CI = 0.03–0.08, 0.09% (95% CI = 0.08–0.12) and 0.14% (95% CI = 0.14–0.15), for individuals infected with Omicron, Delta, individuals who tested negative or who were untested, respectively.

**Risk of post-covid complaints, from 14 to 126 days after test date**
Individuals infected with Omicron had similar rates of musculoskeletal pain, fatigue, cough, heart palpitations, anxiety/depression and brain fog as well as of having any complaint, as individuals infected with Delta, yet had a lower rate of shortness of breath (HR = 0.77, 95% CI = 0.60–0.97) (Fig. 2).

There was a 20–30% increased rate of post-covid fatigue, and 30–80% increased rate of post-covid shortness of breath, both for individuals with Omicron infection and for individuals with Delta infection, when compared to individuals testing negative (Fig. 3) (fatigue: HR = 1.29 (CI: 1.18–1.40) and HR = 1.24 (CI: 1.16–1.33), shortness of breath: HR = 1.29, CI, 1.04–1.61 and HR = 1.69, CI, 1.46–1.96, respectively).

Crude estimates were somewhat different from adjusted estimates (Supplementary Tables 3, 4), suggesting potential confounding from sociodemographic factors or previous health and healthcare use. However, the estimates that were adjusted vs non-adjusted for the number of vaccine doses were rather similar, implying our findings were independent of vaccination status (Supplementary Tables 3, 4). The results were similar in sensitivity analyses including a comparison group of untested individuals ($N = 1,180,716$, median (interquartile range) age 50 (36–60), women $n = 609,189$ (44%), primary school $n = 279,443$ (23.7%), born in Norway $n = 914,226$ (77%), 2 vaccine doses $n = 967,011$ (82%), median (IQR) previous care visits 0 (0–2)) as

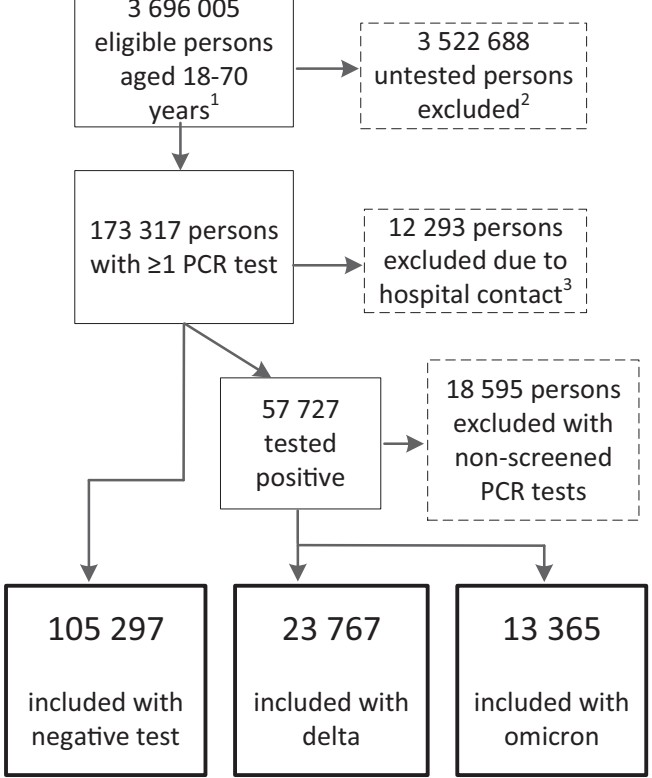

**Fig. 1 | Flow chart presenting eligible, excluded and included individuals in the main analyses.** PCR polymerase chain reaction. Source data are provided as a Source Data file.

Flow chart:

3 696 005 eligible persons aged 18-70 years[1] → 3 522 688 untested persons excluded[2]

173 317 persons with ≥1 PCR test → 12 293 persons excluded due to hospital contact[3]

57 727 tested positive → 18 595 persons excluded with non-screened PCR tests

105 297 included with negative test

23 767 included with delta

13 365 included with omicron

[1] Alive and living in Norway on December 8, 2021. [2] Non-tested or tested for SARS-CoV-2 outside study period (of which N=1 180 716 untested persons were included in sensitivity analyses). [3] Inpatient or outpatient, from -2 to +14 days from test date.

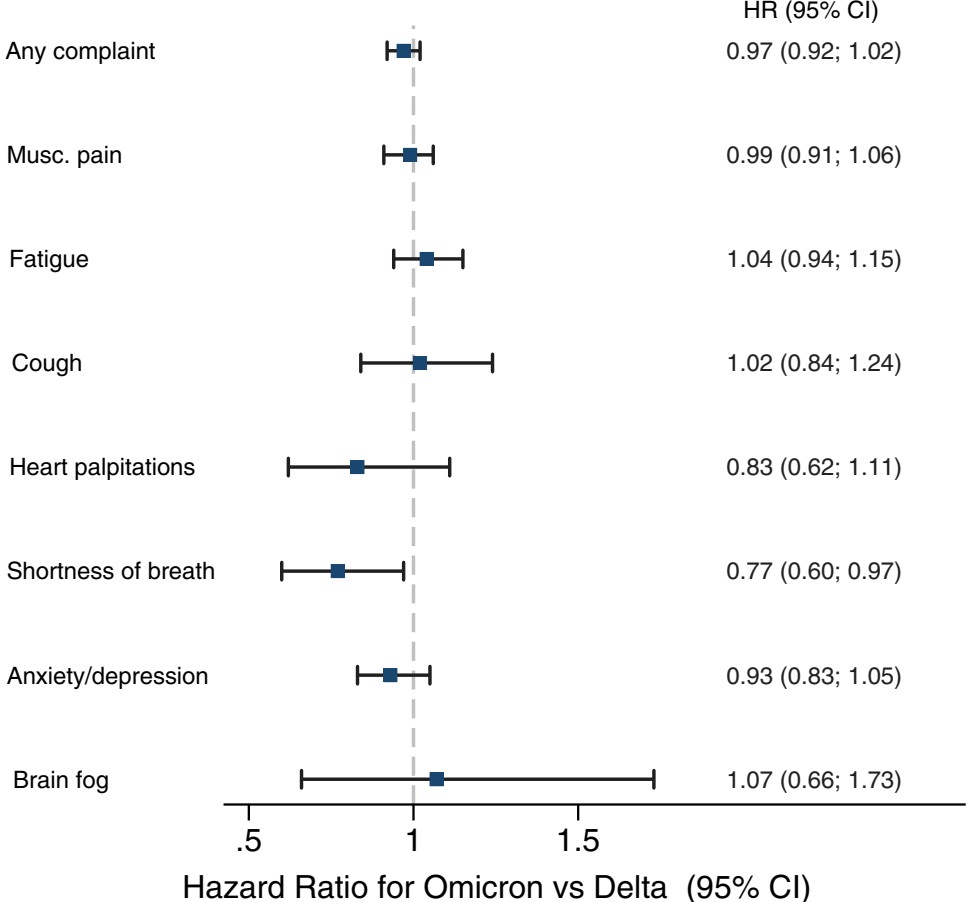

**Fig. 2 | Risks of complaints from 14 to up to 126 days after SARS-CoV-2 infection with the Omicron variant (n = 13,365), adjusted for age, sex, education, comorbidities, test and care activity and vaccination.** Reference category: persons with SARS-CoV-2 Delta (dashed vertical line) (n = 23,767). Data are presented as Hazard Ratios (HR) with 95% confidence interval (CI) and examined over 8 independent experiments, one for each post-covid outcomes. Blue squares represent the estimates for the Omicron variant compared to the Delta variant. Musc. pain=musculoskeletal pain. Source data are provided as a Source Data file.

comparison group (Supplementary Table 2, Supplementary Fig. 2). Estimates were higher, yet of similar magnitude for Omicron and Delta in sensitivity analyses with censoring of observations from the date of positive test in the comparison group (consisting of both persons testing negative and untested persons) (Supplementary Fig. 3). For example, having Delta or Omicron infection was associated with a 20–50% increased risk of cough, heart palpitations and brain fog when compared to individuals testing negative and the untested whose observations were censored from their date of positive test (Supplementary Fig. 3).

**Proportions having post-covid complaints in the different post-covid periods**

The adjusted prevalence of post-covid complaints ranged from 5 to 250 per 10,000 individuals and was generally higher for persons included in the main analyses (Omicron, Delta and individuals testing negative) than for individuals included in the sensitivity analyses (untested individuals) (Fig. 4). In direct comparison of specific complaints following Omicron infection vs Delta infection for the whole post-covid period (14 to up to 126 days), no group differences could be observed for any outcome (Table 2).

However, a few important group differences could be observed after 90 days. First, the analyses of any of the complaints implied that 43 (95% CI = 14–72) fewer individuals per 10,000 individuals infected with Omicron would visit the doctor and have any of the complaints at 90 days or more after testing positive compared to 10,000 individuals infected with Delta (Table 2). Further, 23 (2–43) fewer individuals per

10,000 persons would visit the doctor with musculoskeletal pain at 90 days or more following Omicron infection than at 90 days or more following Delta infection (Table 2).

No- or only minor group differences could be observed for the other post-covid complaints in the different post-covid periods (Table 2). The visually inspected group differences between infected and non-infected slightly increased in sensitivity analyses with censoring of observations from the date of positive test and onwards (Supplementary Fig. 4).

Sensitivity analyses of any complaint stratified on vaccination showed minor group differences up to 90 days after positive test (Supplementary Fig. 5, Supplementary Table 5). However, after 90 days, individuals with Omicron infection who were not vaccinated with their last dose (1, 2 or 3) at 14–210 days before their inclusion (test) date (N = 3997 (29.9%)) would have 81 (33–129) per 10,000 fewer cases with any post-covid complaint compared with individuals with Delta infection with similar no such vaccination (N = 9607 (40.4%)). Among vaccinated individuals (1, 2 or 3 at 14–210 days before their inclusion (test) date), individuals with Omicron infection (N = 9368 (70.1%) would have 36 (1–70) per 10,000 fewer cases with any post-covid complaint at 90 days or more, compared with individuals with Delta infection (N = 14,160 (59.6%)).

## Discussion

In this population-based prospective cohort study, we found that individuals infected with Omicron had similar risk of a range of specific post-covid complaints as individuals infected with Delta, both in the

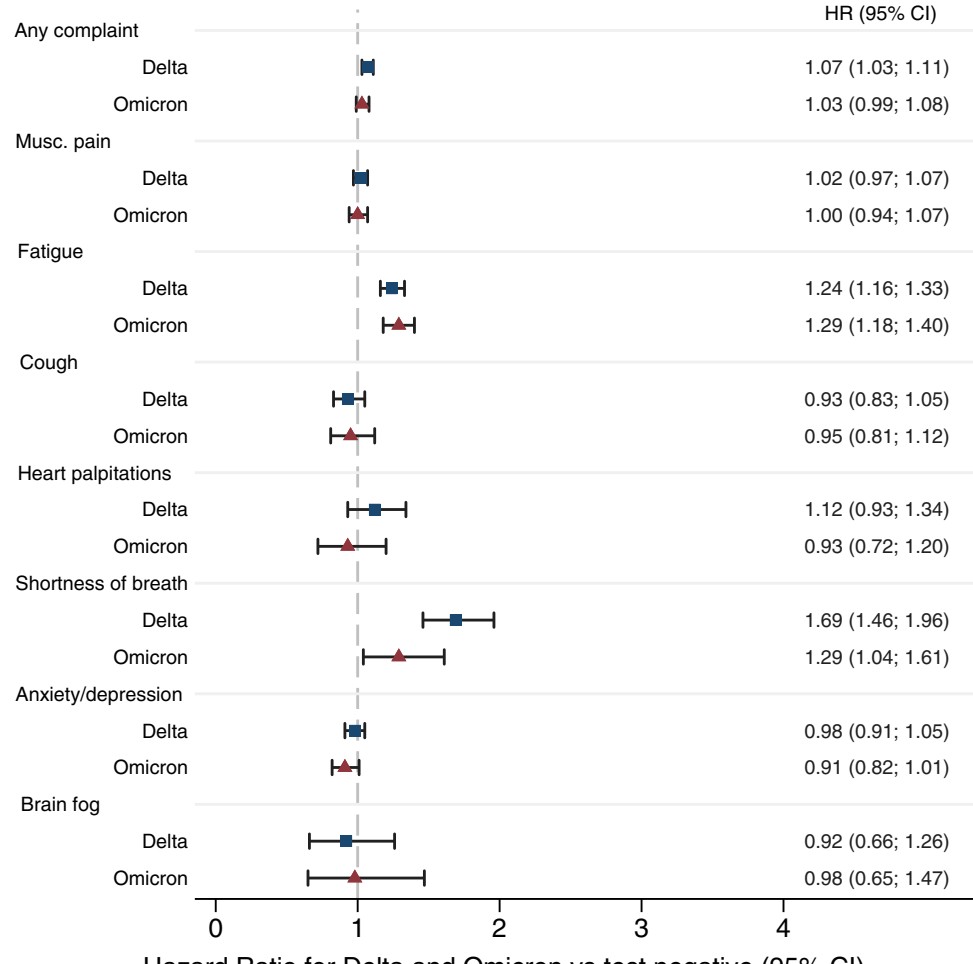

**Fig. 3 | Risks of complaints from 14 to up to 126 days after SARS-CoV-2 infection with the Omicron variant (n = 13,365) and after infection with the Delta variant (n = 23,767), adjusted for age, sex, education, comorbidities, test and care activity and vaccination.** Reference category: persons testing negative (dashed vertical line) (n = 105,297). Data are presented as Hazard Ratios (HR) with 95% confidence interval (CI) and examined over 8 independent experiments, one for each post-covid outcomes. Blue squares represent the estimates for the Delta variant compared to individuals testing negative. Red triangles represent the estimates for the Omicron variant compared to individuals testing negative. Musc. pain=musculoskeletal pain. Source data are provided as a Source Data file.

acute (14–29 days), sub-acute (30–89 days) and chronic post-covid (≥90 days) phases. However, at 90 days or more after testing positive, individuals with Omicron infection had lower risk of having any complaint (43 (95%CI = 14–72) fewer per 10,000), as well as lower risk of musculoskeletal pain (23 (95% CI = 2–43) fewer per 10,000) than individuals with Delta infection.

**Comparison to previous studies**
This is to our awareness the first study to provide estimates of sequelae of the Omicron variant. As such, it sheds new and important light on the growing body of evidence suggesting that Omicron leads to milder acute disease and fewer hospitalizations than Delta[4]. We found no study of post-covid medical records in primary care for an effective comparison of findings to previous studies on other SARS-CoV-2 variants. Thus, the finding of similar prevalence of specific post-covid complaints except for less musculoskeletal complaints after 90 days among Omicron-infected than among Delta-infected has not been previously reported. Several studies have been performed comparing individuals with confirmed COVID-19 (usually for the earliest or non-specified SARS-CoV-2 variants), to individuals with no confirmed COVID-19[10]. We could shed new light on these studies by comparing both the assumed milder Omicron variant and the assumed more severe Delta variant, to similar comparison groups consisting of non-infected as in previous studies. Like previous studies, for example

based on self-reported data or hospital data[1,11], we found that the risk of fatigue and shortness of breath was elevated for the infected compared to the non-infected. Here, we could show that the elevation of risk probably is irrespective of the latest SARS-CoV-2 variant Omicron and Delta (in primary care data).

The Omicron- and Delta-risk of fatigue and shortness of breath relative to non-infected were the highest in the acute and sub-acute post-covid periods, with no difference after 90 days or more (Fig. 4, Supplementary Fig. 4). However, when Omicron and Delta were compared to each other, we found small discrepancies in estimates of shortness of breath from Cox vs logit regression models. In the Cox models, there seemed to be a ~20% decreased risk of shortness of breath in the 14–126-days interval for Omicron vs Delta (Fig. 1). In contrast, the logit models showed no difference in absolute risk for the same period (−3 (−7 to 1) cases per 10,000 with Omicron vs Delta). The different findings in the different models may be due to the Cox model including only the first mention of medical record, whereas the logit models include all records (averaged to 1 per week). Thus, the Cox model would systematically pick the earliest record of shortness of breath, which we find from Fig. 4, Supplementary Fig. 4 and Table 2 are clearly elevated the nearer they come to the test date. It is possible that individuals with shortness of breath (or with other complaints) refrained from contacting the physician for a second time, or that the physician did not bother recoding the same complaint, potentially

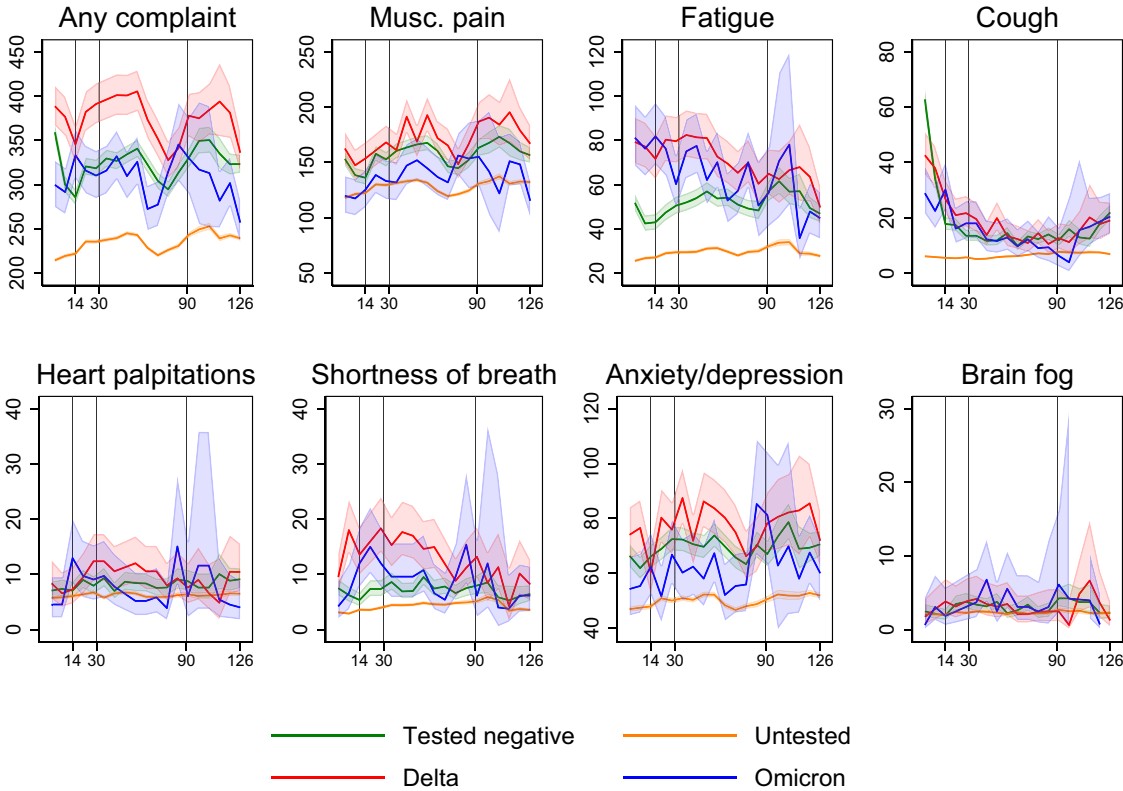

**Fig. 4 | Weekly proportions having post-covid complaints per 10 000 individuals in the acute (14–29 days), sub-acute (30–89 days) and chronic (≥90 days) post-covid phases, as distinguished by vertical lines for days 14, 30 and 90.** Data are presented as the number of individuals visiting primary care for the outcome in question at least once a week per 10,000 individuals in each group (coloured lines), with 95% confidence intervals (shaded area). Estimates are predicted probabilities from a logit model with standard errors clustered on person level, adjusted for age, sex, education, comorbidities, test and care activity and vaccination. Source data are provided as a Source Data file.

**Table 2 | The differences between the group being infected with Omicron and the group being infected with Delta in prevalence of different post-covid complaints over time**

|  | Acute COVID-19 1–13 days | Acute post-covid period 14–29 days | Sub-acute post-covid period 30–89 days | Chronic post-covid period 90 days and more | Whole post-covid period 14 to up to 126 days |
|---|---|---|---|---|---|
| Any complaint | −49 | −4 | −20 | −43 | −23 |
|  | −83 to −15 | −36 to 28 | −46 to 5 | −72 to −14 | −46 to −1 |
| Musculoskeletal pain | −18 | −7 | −2 | −23 | −9 |
|  | −40 to 4 | −28 to 14 | −19 to 15 | −43 to −2 | −24 to 6 |
| Fatigue | 6 | 1 | −3 | −11 | −4 |
|  | −10 to 22 | −13 to 16 | −14 to 7 | −22 to 0 | −13 to 5 |
| Cough | −10 | 3 | 1 | 4 | 2 |
|  | −21 to 1 | −5 to 11 | −4 to 5 | −2 to 11 | −2 to 6 |
| Heart palpitations | −3 | 1 | −4 | −4 | −3 |
|  | −8 to 1 | −4 to 6 | −8 to −1 | −8 to 0 | −6 to 0 |
| Shortness of breath | −8 | 0 | −3 | −3 | −3 |
|  | −14 to −3 | −7 to 6 | −8 to 1 | −8 to 1 | −7 to 1 |
| Anxiety/depression | −15 | −2 | −9 | −5 | −7 |
|  | −32 to 1 | −18 to 14 | −22 to 4 | −20 to 10 | −19 to 5 |
| Brain fog | −1 | −1 | 1 | −1 | 0 |
|  | −3 to 2 | −4 to 1 | −2 to 3 | −3 to 1 | −2 to 2 |

Estimates are group differences in prevalence per 10,000 persons in the respective groups, with 95% confidence intervals, representing the group testing positive with Omicron minus the group testing positive with Delta.

leading to misclassification of complaints towards the later study periods. However, unless the widespread talk of long-covid leads to behavioural responses only among the infected, we would expect such time-differential misclassification to affect all study groups to an equal extent, i.e. it would have limited impact on our findings. Misclassification bias is a common threat to validity in all register-based research and in exchange, such research may provide a good overview of the health service burden posed by a disease.

### Relevance to public health, clinic and future research

Overall, our findings suggest that the included post-covid complaints exist to a similar extent after infection with Omicron as after infection with Delta, at least for the acute and sub-acute post-chronic phases. However, we found indications that the Omicron variant might be milder than the Delta variant at 90 days after testing positive and beyond, in studies of any complaint and in studies of musculoskeletal pain. No group differences after 90 days could be observed for the assumed main post-covid complaints (fatigue and respiratory complaints) as recently defined by the World Health Organization[12] ("persistent complaints, typically fatigue and shortness of breath, with unknown cause still present at 3 months from the onset"). Our findings suggest that Omicron and Delta will lead to a similar burden of such WHO-defined post-covid fatigue and shortness of breath in the long run, yet that there may be fewer visits with any post-covid complaint and fewer visits with musculoskeletal pain in the Omicron-infected than in the Delta-infected.

Thus, our findings may have some important combined clinical and public health messages. First, we provide insights into the natural medical history after infection with Omicron vs Delta in a population where the majority is vaccinated, demonstrating the need to further study the onset, duration and severity of post-covid complaints following the Omicron variant, e.g. using patient-reported or clinical data. We also provide data material to be used by public health workers, policy makers and clinicians, for example for cost estimation and prioritizing resources in primary care in times or regions where many are infected with Omicron simultaneously.

### Strengths and limitations

Strengths of our study include the use of sequenced data allowing for comparison of Omicron vs Delta during the same calendar period when the two variants had the largest overlap, combined with healthcare register data with no attrition. Further, equal access to SARS-CoV-2 testing at no cost for the individuals as well as a universal tax-funded healthcare system, improve generalizability of findings to other countries.

A limitation of our study is that we could not include antigen or home tests as they were not registered. Polymerase chain reaction testing was however mandatory for everyone with a positive antigen test in the beginning of our study period. Moreover, all participants in our study had a PCR test in a period characterized by great uncertainty regarding the severity of the new SARS-CoV-2 Omicron variant. It is possible that our population consisted of particularly health-conscious persons who were highly prone to get tested and who were more prone to seek medical care after knowing they had been ill. Indeed, there were some important differences in baseline characteristics on seeking medical care (testing and healthcare use) and mortality that may impact on our findings through selection/collider stratification and/or confounder bias. We believe our methodological approach ensuring comparison of individuals who were tested in the same calendar week, the inclusion of untested and untested + test negative in sensitivity analyses, as well as the adjustment for a range of covariates including health-seeking behaviour would limit these potential biases. Further, any differential mortality is unlikely to impact on our findings as it was below 0.2% for all study groups.

A second limitation may be that the 10–18% who tested positive after being included with a negative test or no test were unrepresentative to the source population, introducing differential loss to follow-up. More specifically, knowing that infection with the Omicron variant comprised 80% of individuals on December 31, 2021 (Supplementary Fig. 1), rising even further into January 2022, we would also know that close to all individuals who tested positive after testing negative, tested positive with the Omicron variant, further strengthening the dependent loss to follow-up. Further, with the knowledge that (1) Omicron infection is known to result in a milder initial disease course than previous variants[4], probably resulting in less anxiety and less testing in the population, (2) mass vaccination with the third dose mRNA vaccine occurred in Norway begin January 2022[13], probably resulting in fewer tests and, (3) test criteria became less strict throughout the follow-up period[14], embracing fewer and hence resulting in fewer PCR tests but more home/antigen tests (we had no access to test results from home/antigen tests), we can infer that only the most severe Omicron cases with some specific characteristics would have PCR test in place of or in addition to an antigen test during the follow-up period. Censoring these individuals from their date of positive test might violate the assumption of independent censoring, as a participant could be lost to follow-up because one of the outcomes was about to occur. Because our main study aim was comparing Omicron and Delta, for which censoring at positive test was not an issue, we chose to present the proportions non-infected becoming infected during follow-up, as well as conducting analyses with and without censoring of observations from their date of positive test and onwards[15]. As expected, the estimates from analyses with censoring at positive test were higher than estimates from analyses without such censoring (Fig. 3 vs Supplementary Fig. 3 and Fig. 4 vs Supplementary Fig. 4). We believe the alternatives to handle dependent censoring, e.g. imputation[16] or inverse probability weighting[17] would add unnecessary complexity to our study without contributing to responding to our main research question.

A third limitation may be misclassified and potentially underreported and/or underpowered (outcome) data, as briefly described above. For example, we had few observations of brain fog, and estimates should be interpreted with care. To face these challenges, we added an outcome including any of the specific complaints, consistently showing in the main and sensitivity analysis (stratified by vaccination status) that there might be greater differences between Omicron and Delta than found in analyses of each of the specific complaints. Interestingly, the largest group differences were seen for the chronic post-covid period, with absolute risk difference magnitudes −43 (−72 to −14) per 10,000 for the whole cohort and −81 (−129 to −33) per 10,000 for unvaccinated and −36 (−70 to −1) per 10,000 for vaccinated. We believe these findings suggesting Omicron is similar to Delta in the acute and sub-acute post-covid phase, but milder than Delta in the chronic post-covid phase warrant more investigation in future studies with longer follow-up periods. Further, the study of vaccination against COVID-19 and post-covid complaints is complex due to potential collider bias and healthy vaccinee bias[18]. Our findings by strata of vaccination can only be regarded as explorative and should be confirmed using more suitable methods for causal inference from observational designs. For example, future studies could look into the effects on symptom reporting post vaccination. A final limitation may be preferential sequencing of suspected Omicron samples over Delta. If present, we believe it had limited impact on our findings, as Supplementary Fig. 1 shows that the inclusion period covered the period with the greatest overlap between the variants (50-50 around December 24, 2021).

In conclusion, we found that individuals infected with Omicron had similar risk of a range of specific post-covid complaints (fatigue, cough, heart palpitations, shortness of breath and anxiety/depression) as individuals infected with Delta, both in the acute (14–29 days), subacute (30–89 days) and chronic post-covid (≥90 days) phases. However, at 90 days or more after testing positive, individuals infected with Omicron had lower risk of having any complaint as well as lower risk of having musculoskeletal pain than individuals infected with Delta.

## Methods

### Design and data sources

Using a prospective cohort study design applied to data in the Norwegian Emergency Preparedness Register (S-Table 1)[19], we included all

**Table 3 | Condition/complaint with corresponding diagnostic code (ICPC-2) used in primary care[a]**

| | |
|---|---|
| Pain (general/multisite and localized pain and symptoms from the musculoskeletal system, not classified elsewhere (neck, back, arms/hands, feet/legs)) | A01, L01–L17, L18–L20, L29 |
| Fatigue | A04, A05, A29 |
| Cough | R05 |
| Heart palpitations | K04, K05, K29 |
| Shortness of breath | R02 |
| Anxiety and depression | P03, P76, P01, P74 |
| Brain fog (concentration or memory problems) | P20 |

[a]With condition/complaint we refer to all information that may be included in an ICPC-2 (International Classification of Primary Care 2) code: Diseases, disorders, signs, symptoms and/or complaints as classified by the physician consulted.

Norwegian residents aged 18–70 years, who tested negative or positive for SARS-CoV-2 with known variant during the period that the Omicron and Delta variants had the greatest overlap in Norway; from December 8 to December 31, 2021. These data were linked on the personal ID number to provide information on healthcare contacts in primary care (general practitioners and emergency wards) with specific medical record (Supplementary Table 1). The PCR testing criteria were constant throughout the study period and included persons with symptoms of COVID-19, persons in close contact with anyone with COVID-19 as well as persons having a positive antigen test. Screening for SARS-CoV-2 variant was performed by Sanger or whole genome sequencing on all positive PCR tests if the laboratories had capacity and only on positive tests with suspected Omicron if the laboratory had capacity challenges. We excluded all individuals with previous positive PCR tests (up until December 7, 2021, to avoid pre-existing post-covid complaints), individuals with unscreened positive tests and all individuals who had a hospital contact from −2 to +14 days from the test date[8]. In this way, we could study individuals with known infection with SARS-CoV-2 Omicron and Delta variant with assumed mild disease courses and/or who were known not to be tested as part of hospital contact or routine testing at hospitals. Participants were categorized into three study groups based on their test result and date of testing: (1) individuals infected with Omicron, (2) individuals infected with Delta and (3) individuals who were non-infected (tested negative during the study period and/or earlier but allowed to test positive after the test date).

## Outcomes

The outcomes included were the most frequently reported post-covid complaints in systematic reviews[10] (Table 3), as registered in medical records with high validity and reliability[20] from day 14 after positive test, and onwards: musculoskeletal pain, fatigue, cough, heart palpitations, shortness of breath, anxiety/depression and brain fog, as well as any of the complaints. We allowed for having multiple complaints. We assumed no competing risk between outcomes, i.e. having a record with e.g. fatigue was assumed not to preclude having a record with e.g. cough.

## Statistical analyses

First, we described the study groups on baseline and follow-up characteristics using means with standard deviations, numbers observed with proportions and proportions with 95% confidence intervals based on Wilson[21]. Second, we calculated the person-time (numbers of included persons multiplied by their number of days from the test date to their date of censoring) with the number of failures (the outcome in question) and incidence rate with 95% confidence interval for all study groups and all outcomes. If an individual had multiple records with the same complaint within the follow-up period of interest (or

combination of diagnostic codes indicative of the complaint, as categorized in Table 3), we chose the first one. At least one day of follow-up was required and observations were censored at day 126, date of death or emigration, whichever came first. Individuals who were included with a negative test and later tested positive were not censored from their date of positive test in the main analyses as it would violate the assumption of independent censoring[15] (the Omicron variant was dominant in Norway from December 24, 2022 (S-Fig. 1), i.e. the majority of positive tests during the follow-up period would be caused by the Omicron variant).

Third, and based on these person-time and failure data, we estimated the hazard ratio (HR) with 95% confidence intervals (CI) for having the potential post-covid related complaints/diagnoses in primary care, from 14 to up to 126 days after the test date using Cox regression analyses unadjusted and adjusted for age, sex, education level (no education to >1 year college/university education in four categories), the number of comorbidities in 2020-21 (0–1 vs 2 or more)[22], the number of previous negative tests in 2020-21 (0, 1, 2 or 3 more) and the number of previous all-cause primary care visits in 2020-21 (0–10 or more) as potential confounders. We checked specifically for potential confounding by vaccination status (the number of mRNA COVID-19 vaccine doses: 0, 1, 2 or 3 or more). All data on potential confounders were identified in the same data sources, i.e. in National register data covering the entire Norwegian population from January 1, 2020 (only education status was registered in 2019) (Supplementary Table 1).

Standard errors were calculated using bootstrapping (50 replications). The Delta variant dominated earlier than Omicron (S-Fig. 1), thus we expected differences in follow-up time by variant. This was handled by stratification on the test week in the regression models. The stratification ensured that there was no possible non-proportionality of hazards resulting from the test week (although there may be violation of the assumption of proportional hazards for other variables). In this way, i.e. by choosing the time period where the overlap between the two variants was the greatest, and by stratifying on the calendar week of testing, we could limit any potential bias arising from potentially differential temporal trends in test and screening patterns by variant. Still, the hazard ratio estimate from a Cox proportional hazards model should only be regarded as a weighted average of the time-varying hazard ratios, i.e. a summary of the treatment effect during the follow-up[23].

Thus, to assess whether the post-covid complaints were more or less common in certain periods after positive test (the acute phase (14–29 days), the sub-acute phase (30–89 days) and the assumed chronic post-covid condition phase (90–126 days) as recommended in previous studies[12,24], we estimated the group-wise weekly proportions having the outcome in question (with 95% CI) and plotted the predicted probabilities from a logit model with standard errors clustered on person level, adjusted for the same covariates as described above. In these analyses, all medical records were included (i.e. not only the first one as for the Cox regression analyses—rather, all visits every week were included and dichotomized into having the outcome in question that week, yes or no). For each post-covid phase, we also calculated the group difference in prevalence for individuals infected with Omicron vs individuals infected with Delta, by subtracting the estimate for individuals with Omicron infection from the estimate for individuals with Delta infection.

Finally, we performed several sensitivity analyses. First, because individuals testing negative may be more prone to get tested and subsequently visit primary care due to (persistent) symptoms from similar bodily systems as those affected by SARS-CoV-2, we repeated the main and time-differentiated analyses using a comparison group consisting of the untested individuals (aged 18–70 years, non-hospitalized and assigned a random, hypothetical test date during our study period) in a sensitivity analysis. The untested individuals

were included from their randomly assigned test date and were never tested prior to this date (they were allowed to have positive tests after the inclusion date but not required to). Second, to assess the impact of positive PCR tests during the follow-up time among the non-infected (individuals testing negative and the untested, joined to one comparison group), we repeated the analyses with censoring of observations from their date of positive PCR test and onwards (allowing for dependent censoring, i.e. assessing potential differential loss to follow-up under different assumptions)[15]. And finally, because vaccination and time from vaccination may affect our findings, we repeated the time-differentiated analyses by stratifying the logit model on vaccination status and time since vaccination (having received the latest dose (1st, 2nd or 3rd dose) in the time interval 14–210 days prior to the inclusion/test date, yes or no, i.e. similar categorization as in our recent study on vaccination and medical complaints[18]) among the infected. Because of few observations and a likely low statistical power in such stratified analyses, these analyses were only performed for the outcome including any of the symptoms. All analyses were run in STATA MP v. 17.

### Patient and public involvement
No patients were involved in setting the research question, study design, outcome measures, or the conduct of the study. The study was based on deidentified data from Norwegian national registries.

### Inclusion and ethics
The Ethics Committee of South-East Norway confirmed (June 4, 2020, #153204) that external ethical board review was not required. The data sources (The emergency preparedness register for COVID-19 (Beredt C19)) and methods used were regarded by the ethical committee to respond to research aims not falling within the Law of Health Research §§ 2 and 4a. Their resolution was also based on the fact that the data sources were established and handled in accordance with the Health Preparedness Act §2-4 (11), enabling a quick and responsive way for the Norwegian government to access knowledge of how to handle the pandemic. No informed consent from participants was required since our study was based on routinely collected register data covering the entire Norwegian population. Data from the different registers included in the study were linked by the responsible researchers using an encrypted personal ID-variable. The researchers responsible for the data linkage and analyses had no access to the unencrypted ID-numbers. All methods were carried out in accordance with relevant guidelines and regulations. To protect participants privacy and security of personal data, all data were handled under strict confidentiality and access control as described in the Norwegian Institute of Public Health's internal documentation.

### Reporting summary
Further information on research design is available in the Nature Portfolio Reporting Summary linked to this article.

## Data availability
The dataset of this study was the Emergency Preparedness Register for COVID-19 (Beredt C19), which is a property of the Norwegian Institute of Public Health that was provided to the researchers through a restricted-access agreement that prevents sharing the dataset with a third party or publicly. The raw Beredt C19 data are protected and are not available due to privacy law. Thus, individual-level data of patients included in this paper after de-identification are considered sensitive and will not be shared. However, the individual-level data in the registries compiled in Beredt C19 are accessible to authorized researchers after ethical approval and application to "helsedata.no/en" administered by the Norwegian Directorate of eHealth. Data requests may be sent to "service@helsedata.no. Source data are provided with this paper.

## Code availability
All computer codes used to analyze the data relevant in this study were written and run in STATA MP v. 17. The custom code and description of the code's functionality is provided in text (starred) in the attached.txt file with file name "Supplementary Code 1".

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

## Acknowledgements

This study was supported by the Foundation for Research in Rheumatology (FOREUM—grant number 054), and by the Norwegian Institute of Public Health (internal funding, no grant number available). The funding sources had neither influence on the design or conduct of the study; collection, management, analysis, nor the interpretation of the data; preparation, review or approval of the manuscript; nor the decision to submit the manuscript for publication. We thank the Norwegian Directorate of Health, particularly Olav Isak Sjøflot and his Department of Health Registries for their cooperation in establishing the emergency preparedness register, and Gutorm Høgåsen and Anja Elsrud Schou Lindman for their invaluable work on the register. The interpretation and reporting of the data are the sole responsibility of the authors, and no endorsement by the register is intended or should be inferred. We also thank staff at the Norwegian Institute of Public Health who have been part of the outbreak investigation and response team.

## Author contributions

K.M. had full access to all of the data in the study and take responsibility for the integrity of the data and the accuracy of the data analysis. K.M. drafted the manuscript. K.M. and J.M.K. contributed with concept and design. K.M., J.K.M. and A.T. contributed with statistical analyses. M.E. obtained funding. All authors (K.M., D.T.K., A.D.I., A.K., A.T., J.R., S.B.-Z., M.E., P.M.M. and J.M.K.) contributed with acquisition or interpretation of data. All authors (K.M., D.T.K., A.D.I., A.K., A.T., J.R., S.B.-Z., M.E., P.M.M. and J.M.K.) critically reviewed the manuscript for important intellectual content. All authors (K.M., D.T.K., A.D.I., A.K., A.T., J.R., S.B.-Z., M.E., P.M.M. and J.M.K.) gave final approval for the version to be submitted.

## Competing interests

M.E. reported grants from The Swedish Research Council, grants from Österlund Foundation, grants from Governmental Funding of Clinical Research within National Health Service (ALF), grants from Greta and Johan Kock Foundations, grants from The Swedish Rheumatism Association, during the conduct of the study. The remaining authors declare no competing interests.
