## [Peer Review File · Nature Communications]

Post-covid medical complaints following infection with SARS-CoV-2 Omicron vs Delta variantsREVIEWER COMMENTS

Reviewer #1 (Remarks to the Author):

This is an interesting and informative paper showing increased reporting of fatigue and shortness of breath for up to 126 days (censor period) following documented SARS-CoV-2 infection, based on linkage to population primary care and infection data in Norway.

The authors had access to sequencing data so they were able to compare symptoms following Delta and Omicron infections (presumably BA.1 given the timing although this is not stated). As the timing of the Delta and Omicron waves differed (Omicron rapidly replacing Delta) the authors constrained their analyses to a 3+ week period in December 2021 when both variants were circulating.

There are a number of issues the authors should consider:

Major:

More detail is required of the statistical method including clarifying what is meant by “the person-time with the number of failures (the outcome in question)” (notwithstanding the footnote to Table 3 which is a bit cryptic) and the use of a Cox model. Likewise the meaning of “failures” needs to be better described in footnote Table 3 and added to Suppl Tables 6-8. Concerning Long COVID there is not only interest in persistence of symptoms, but loss of symptoms and when that occurs, and whether symptoms relapse. This would be a valuable addition to the paper. In trying to capture the potential relapsing-remitting nature of Long COVID, you might consider e.g. using a model with finer stratified periods of follow-up, where outcome is yes/no to symptom(s) being present at some time during that period. Using only first mention of symptom(s) in the Cox model, and ignoring previous occurrence of symptoms for the three periods (14-30 days, 30-90, 90-126 days), do not allow for this possibility. (Incidentally, the periods of follow-up should be non-overlapping.)

Is there any adjustment for previous SARS-CoV-2 infection? The main model adjusts on number of previous negative SARS-CoV-2 tests, presumably to control for test-seeking and other behaviours, but it would be important to account for the possibility of differential infection-driven immunity on symptom reporting.

Also it would be of interest to see whether symptom reporting post-infection is modified by subsequent vaccination – are there sufficient data to address that question? (It has been suggested that one reason for persistence of symptoms is persistent low-level infection which may be modified by vaccination.)

There is particular interest in symptomatic (‘breakthrough’) infection among the fully vaccinated population and it would be of interest (sensitivity analysis) to stratify by vaccine status (i.e. number of doses) rather than just relying on adjustment in the models. It would also be important to adjust on time since most recent vaccination.

Again in sensitivity analysis, you could match Delta and Omicron cases on at least age and sex (not just week) given apparent differences between these groups.

“The mortality during follow-up was low (0.08% (95% CI=0.06-0.10), 0.03% (95% CI=0.01-0.06) and 0.04% (95% CI=0.00-0.06) for persons testing negative or positive with delta and omicron, respectively” – Is this correct? Although CIs are overlapping (just) it does seem that the test negative group may have higher mortality implying this group may be different to the others in ways that have not been captured (shielding for example). They also appear to have higher reporting of anxiety/depression than the Delta or Omicron groups. This should be discussed.

Minor:

I would include supplementary Fig 2 in the main part of the paper. This says 20-70 years, the text says 18-70 years.

There is a hint in the conditional logit model (Suppl Table 5) that fatigue reporting is higher for Delta than Omicron (different to the continuous Cox model) – this should be commented on.

What is the size of the untested group in Suppl Table 9?

In limitations, there should be mention of potential bias from preferential sequencing of suspected Omicron samples over Delta.

This is not the only study to look at post-Omicron persistent symptoms, and other studies suggest persistence beyond 90 days which will be a continuing burden on health services (“this burden will cease when 90 days have passed”) – introduction and discussion need updating accordingly.

The authors might refer to Whitaker M et al Nat Commun 2022, Apr 12;13(1):1957 which identified two clusters of symptoms post-SARS-CoV-2 infection for earlier variants, one characterised by fatigue and one by shortness of breath.

“novel insights into disease etiology of post-omicron” – this is not really looking at etiology rather natural history post infection?

Abstract: “Omicron was related with a similar, and no increased risk of musculoskeletal pain, cough, heart palpitations, anxiety/depression when compared to delta and when compared to test negative” – confusing wording, re-word “Compared with either test negative or Delta, people testing positive for Omicron did not have increased...”

“sanger” should be “Sanger”

“covid” should be “COVID-19”

“Moreover, all participants in our study had a PCR” – typo for participants.

Omicron and Delta should have capitals throughout.

Reviewer #2 (Remarks to the Author):

This is an interesting and timely paper as there has been much made in policy terms that ‘Omicron is milder’ and therefore causes less Long Covid. The effect of Omicron compared to Delta in acute COVID is heavily confounded by vaccination and prior infection, and not always in a positive way as the recent paper by Boynton and Altman shows. Given that Long Covid is starting to become the lasting legacy of the pandemic with 100,000s of long term disabled in many countries, whether Omicron, and presumably its various sub strains is less likely to cause Long Covid is an important issue for policy (or at least it should be).

What is important here is that this is a controlled observational cohort comparing, in the same space of time, PCR negative, PCR+ve Delta and PCR+ve Omicron, with follow up up to 126 days using routine primary care electronic health record data. However, there are some issues, and these relate to power and to potential ascertainment bias in the outcome measure (failure = one of a set of ICPC2 codes for recognised Long Covid symptoms being recorded by the GP). Lets unpick these in turn:

- Power issues. We don't have a section in the paper on power, but its important as the population is

being split in three, then into three time periods, then into 7 sets of sub-symptoms, then there are those without tests available, or without data available. Although the analysis is based on person-days, there are 242,264 person days in the Omicron group at 90-126 days (supplementary table 8), assuming a full 126 days for each, that is less than 2,000 individuals - or at 5% (being current accepted estimate of Long Covid risk at 6 months) that would be 100 people split up between the 7 symptom sets. Its easy to see that all these sub group analyses are really underpowered. A rule of thumb would be 50 individuals minimum per cell. I don't think this issue is adequately addressed in the paper and I think the authors need to do two things 1. Do some post-hoc power calculations (correcting also for the multiple comparisons) 2. Restrict the main findings to those cells that are adequately powered, regardless of the result significance and recognise that all others are 'exploratory only'.

- Ascertainment bias. Although in the Norwegian payment system GPs need to put at least one ICPC2 code into a consultation, there is by no means comprehensive recording of ALL symptoms present when a patient consults. Only coded data are analysed, not free text, so we are almost certainly hugely underestimating the actual prevalence of symptoms in this group of patients. This explains a very large part of the gap between incidence rates reported here and in patient-reported surveys. In addition I looked at REF 12 in detail, which is used to justify the statement that the GP records are a reliable source of symptom data. The study is based in practices in a single region who are already active in a research network. Altogether codes were not present in 144 of a total of 398 (36 %) 'simple contacts' (with no issuance of a prescription). Generally ICPC2 is a decent way of capturing symptom codes in primary care, but in Long Covid the use of P20 'memory disturbance' isn't really a good map to brain fog - which is largely a mental tasking problem. There is also a risk of systematic underreporting of data from multimorbid patients, since the KUHR database only imports the first two diagnoses from the reimbursement cards.

One way around these issues is to look at an analysis of 'any Long Covid symptom'. This would still be subject to ascertainment bias but would be better powered.

Overall I have some issues with the interpretation that the authors put on the results in the abstract and discussion. Firstly we are told that "no increased risk of musculoskeletal pain, cough, heart palpitations, anxiety/depression when compared to delta and when compared to test negative" - this is not reliable on account of the power and ascertainment issues. I also don't think the statement 'The omicron variant will likely lead to a temporarily increased burden on healthcare services.' is warranted. Given the power and ascertainment issues we just don't know that Long Covid is 'temporary' - in fact all the other data - especially UK ONS surveys suggest that for about half of patients it is not.

Professor Brendan Delaney
Imperial College London

Reviewer #3 (Remarks to the Author):

Magnusson et al. use Norwegian health registry data to compare the incidence of a set of complaints that commonly occur following Covid infection between individuals who were infected with the Omicron variant, individuals who were infected with the Delta variant and individuals that were tested and found negative.

The study is based on seemingly high-quality data, and there is value in knowing the incidence of "soft complaints" of the type studied here following Covid-19 infection with different variants, but I do have certain concerns.

== Design ==

In the main analysis, the authors condition on being tested. This conditioning has advantages – it seems to lower the probability of exposure misclassification - but also has disadvantages – most of all that it exposes the study to collider stratification bias. This is further supported by the fact that the “tested negative” group had higher mortality during the study period than both infected groups (!), suggesting that testing is a result of both being infected and of things that result in a higher risk of death (=it is a collider). The authors must acknowledge this in the Discussion.

In the same context, the authors wisely perform a sensitivity analysis in which a negative test is not required for the control group. I have two concerns regarding this (important) analysis: first, it would be preferable to ignore testing altogether in this group and include both tested and untested, instead of specifically excluding the tested. Second, even if the current decision to include only untested is kept, the criteria should under no circumstance condition on being tested after the random inclusion date, which would result in clear selection bias. So, in this case, the sentence “never tested for SARS-CoV-2” in the Methods section should only pertain to the pre-study period.

This important study does not address a causal question (i.e., because the exposure is not manipulable, and because the causal contrast is ill-defined for persons who would not be infected by one or both variants at a given exposure level), and the authors wisely avoid causal language. Despite this, the variables adjusted for in the analysis are called confounders, which is a causal term. While this mishap is common in the literature, the phrase could be omitted altogether (perhaps replaced with “covariates”) to avoid any such error.

Outcomes were identified using primary care data only. I am not sure this is reasonable. Would inclusion of specialist care and hospitalizations not improve the accuracy of the outcomes? Is this data available?

I could not understand the design in the important analysis of “time differentiated risks”. If a person was coded with an outcome during an early period, is he allowed to recur in a later period? If the answer is yes, then wouldn’t misclassification be a serious issue (e.g., because the codes persist in the EMR, or alternatively because the physician wouldn’t bother recoding the same problem, or alternatively because the person would not bother approaching his physician again with a not-so-treatable complaint such as “fatigue”). If the answer is no, then “dilution of susceptibles” could fully explain the reduction in hazard rates over time.

I could not find mention of the types of vaccines used. Is this mostly Pfizer? Moderna? mRNA in general? A mix? Given the documented negative association between vaccination and long-Covid rates, this could be an important predictor?

Given the relatively mild associations found (notwithstanding shortness of breath), the authors must acknowledge ascertainment bias as a possible explanation for the differences from the uninfected. Given the wide-spread talk of long-Covid, both the patients and their physicians would be more likely to complain about and document outcomes such as fatigue following a known infection.

A covariate that would be interesting to explore is “time from vaccination”, as we now know that certain facets of immunity wane rather quickly following vaccination. One could stipulate that Delta infections occurred closer to person’s vaccination, which resulted in less post-Covid. This would be interesting as a covariate and also as an interaction term with the infecting variant (the main exposure), as one could hypothesize that time-from-vaccination is more important for the Delta variant, against which the vaccination is more effective in general.

Though I hesitate to offer yet more scientific questions, it would be interesting to try and ascertain the severity of the infection during the exposure period (the first 14 days following diagnosis), and address the known hypothesis that more severe infections = more post-Covid.

Tables 4 and 5 and all similar tables in the supplementary would be more useful as forest plots (like figures 1 and 2) with side-by-side columns for each time period.

== Analysis ==

The main purpose of the study, as suggested by the title and the first line of the abstract, is comparing outcomes following Delta vs. outcomes following Omicron. This would make the information in Figure 2 (contrasting Omicron vs. Delta) the main result of the paper. Despite this, when results are cited in the Abstract and Results section text, the results cited are from Figure 1 (contrasting Omicron and Delta vs. uninfected). The authors should explicitly define their main study question and report the main results accordingly.

In general, one should not report a difference of two parameters without directly contrasting them. In this regard, statements such as “The risk of complaints was the highest in the acute phase (14 to 30 days) and decreased for both variants in the sub-acute (30 to 90 days) and assumed chronic post-covid condition (here: 90 to 126 days) phases” in the Discussion section do not seem well founded without a direct comparison being made and its uncertainty (i.e., confidence intervals) reported.

The authors at times commit the “absence of evidence is not evidence of absence” fallacy, for example when stating “no increased rates of ... and brain fog in any of the post-covid clinical phases” in the Discussion section, when the CI estimated was 0.68-1.86, which is a noisy estimate that is also compatible with a very strong effect (86% increased risk!). The authors should rephrase more cautiously.

The main analysis in the paper consists of Cox proportional hazards models. As executed in this study, these models assume proportional hazards, which are unlikely to (and some say, cannot possibly) be true in real data. The authors should heed the advice of Stensrud et al. (<https://jamanetwork.com/journals/jama/article-abstract/2763185>), accept that average HRs are being reported, and change the analysis accordingly to correct the standard errors. In this context, it should be noted that stratification by test week only “solves” possible non-proportionality that results from the test week. The text is not clear about that.

I could not find mention in the paper of what happens to persons in the “tested negative” group if they are found positive in a different test during the follow-up period. Their data should be censored, if that were not done. Regardless, it should be reported.

I could not understand the analysis that was performed with conditional logistic regression. I understood that some sort of matching was done, but between whom? And how was censoring handled in this context? Given that I consider stratification by calendar week as sufficient for the concern of confounding by calendar time, I am not sure this analysis is warranted.

In the first paragraph of the Results section, the authors report the incidence proportion of mortality in the different groups with 95% confidence intervals “calculated based on Wilson”. I do not know who Wilson is, there is no citation attached to this sentence, and in general explanations of methodology belong in the statistical analysis section.

In table 1, median [IQR] for age would be more helpful.

We would like to thank the expert reviewers for having performed a careful review and consideration of our study, which we think has greatly contributed to further improve the quality of our work. Please see the detailed point-to-point responses and actions to the reviewers' comments beneath. Our page/table/figure references refer to the revised version of the manuscript *with* marked changes.

Reviewer 1 comments	Our response	Action
This is an interesting and informative paper showing increased reporting of fatigue and shortness of breath for up to 126 days (censor period) following documented SARS-CoV-2 infection, based on linkage to population primary care and infection data in Norway. The authors had access to sequencing data so they were able to compare symptoms following Delta and Omicron infections (presumably BA.1 given the timing although this is not stated). As the timing of the Delta and Omicron waves differed (Omicron rapidly replacing Delta) the authors constrained their analyses to a 3+ week period in December 2021 when both variants were circulating. There are a number of issues the authors should consider:	Thank you for the summary and encouraging comments.	
Major: More detail is required of the statistical method including clarifying what is meant by “the person-time with the number of failures (the outcome in question)” (notwithstanding the footnote to Table 3 which is a bit cryptic) and the use of a Cox model.	We agree.	We have added to the Methods section, p. 4: “First, we calculated the person-time (numbers of included persons multiplied by their number of days from the test date to their date of censoring) with the number of failures (the outcome in question) and incidence rate with 95% confidence interval for all study groups and all outcomes. If an individual had

		multiple records with the same complaint within the follow-up period of interest (or combination of diagnostic codes indicative of the complaint, as categorized in Table 1), we chose the first one. Second, and based on these person-time and failure data, we estimated the hazard ratio (HR) with 95% confidence intervals (CI) for having the potential post-covid related complaints/diagnoses in primary care, from 14 to up to 126 days after the test date using Cox regression analyses unadjusted and adjusted for age, sex, education level (no education to >1 year college/university education in four categories), the number of comorbidities in 2020-21 (0-1 vs 2 or more)¹³, the number of previous negative tests in 2020-21 (0, 1, 2 or 3 more) and the number of previous all-cause primary care visits in 2020-21 (0 to 10 or more) as potential confounders.” We have revised the footnote of Table 3: “Failures represent the first medical record registered at the general practitioner or emergency ward with the diagnoses in question (musculoskeletal pain, fatigue etc. assuming no competing risk between the different diagnoses), from 14 to up to 126 days after the test date.”
--	--	--

		In addition, we have provided details to this assumption at p. 4: “We assumed no competing risk between outcomes, i.e. having a record with e.g. fatigue was assumed not to preclude having a record with e.g. cough.”
Likewise the meaning of “failures” needs to be better described in footnote Table 3 and added to Suppl Tables 6-8.	We agree.	The footnote in Table 3 has been revised, please see action described to the previous comment. S-Tables 6-8 have been omitted as action to other reviewer comments and the reviewer suggestions regarding these tables are no longer applicable.
Concerning Long COVID there is not only interest in persistence of symptoms, but loss of symptoms and when that occurs, and whether symptoms relapse. This would be a valuable addition to the paper. In trying to capture the potential relapsing-remitting nature of Long COVID, you might consider e.g. using a model with finer stratified periods of follow-up, where outcome is yes/no to symptom(s) being present at some time during that period. Using only first mention of symptom(s) in the Cox model, and ignoring previous occurrence of symptoms for the three periods (14-30 days, 30-90, 90-126 days), do not allow for this possibility. (Incidentally, the periods of follow-up should be non-overlapping.)	Although self-reported data would be better suitable to shed light on onset/loss/relapse of symptoms, we agree it would be possible to explore the relapsing-remitting nature of long-covid by applying a finer model to our register data.	For every week after the inclusion and up until week 19 after the test (day 126), we have provided the group-wise proportions visiting primary care with each of our different outcomes. In these analyses, we included all registered outcomes (i.e. not only the first one - rather, all visits every week were included and dichotomized into having the outcome in question that week, yes or no). The weekly proportions were calculated from a logit model with robust standard errors (clustered on patient), with having the complaint (yes/no) in any of the respective weeks as outcome variables for all our study groups, adjusted for the same potential confounders as in the main Cox regression analyses (please see description of the analyses in the methods section at p. 5-6). Besides plotting the weekly prevalence of each complaint for infected and non-infected, we also estimated the absolute

		group difference in prevalence, for persons with Omicron vs Delta for the different post-covid periods: acute (14-30 days), sub-acute (31-90 days) and chronic (91 days or more). Please see Figure 4, Table 3 and a description of results under sub-heading Time-differentiated shares having post-covid complaints, p. 16. The Cox regression analyses stratified by the different post-covid periods, which used only first mention of symptoms were omitted from the manuscript in order to meet the concerns raised by Reviewer 2 and Reviewer 3 regarding power and interpretability.
Is there any adjustment for previous SARS-CoV-2 infection? The main model adjusts on number of previous negative SARS-CoV-2 tests, presumably to control for test-seeking and other behaviours, but it would be important to account for the possibility of differential infection-driven immunity on symptom reporting.	In order to avoid already pre-existing post-covid complaints in our study population, we excluded all individuals with positive tests prior to our inclusion period. We agree this could have been better communicated in our methods section.	We have added the following to p. 4: “We excluded all persons with previous positive PCR tests (up until December 7th 2021, to avoid pre-existing post-covid complaints), persons with unscreened tests and all persons who had a hospital contact from -2 to +14 days from the test date⁸.”
Also it would be of interest to see whether symptom reporting post-infection is modified by subsequent vaccination – are there sufficient data to address that question? (It has been suggested that one reason for persistence of symptoms is persistent low-level infection which may be modified by vaccination.)	We agree this is an important research question. However, considering the large amount of analyses required to assess the association between SARS-CoV-2 variant and post-covid outcomes properly as performed in the current study, and the large amount of analyses required to assess the association between vaccination and complaints properly	None performed.

	(as recently done in our preprint publication Methi et al., medrxiv 2022) we regard the suggestion to be a separate research question deserving a new research paper.	
There is particular interest in symptomatic ('breakthrough') infection among the fully vaccinated population and it would be of interest (sensitivity analysis) to stratify by vaccine status (i.e. number of doses) rather than just relying on adjustment in the models. It would also be important to adjust on time since most recent vaccination.	We agree infection in the vaccinated population is of interest, however, it is also complicated as it in the current study would imply potential collider stratification bias due to conditioning on already testing positive. In our recent preprint publication using similar outcome data as in the current study, we take specific action to overcome such threats of bias (Methi et al., medrxiv 2022). The study did not include data on SARS-CoV-2 variant due to probable low statistical power. However, with the addition of the outcome "any complaint" in the current paper, and with the update of outcome data and use of finer model (logit model plotting weekly proportions) as suggested by the reviewers, we believe it would be valuable to include an explorative analysis with stratification on vaccine and time since vaccination, in its simplest form, as a sensitivity analyses.	An mRNA vaccine is effective after 14 days and for up to half a year after the injection, with a more prolonged effect for the 2nd and 3rd dose than for the 1st dose (Coronavirus immunization program in Norway). Due to low numbers having received 0 or 1 dose, making it challenging to stratify by number of e.g. number of doses and the time since they were received and still observe a sufficient number of outcomes, we included both presence and timing in our explorative analyses, by stratifying on having received an mRNA vaccine against SARS-CoV-2 in the time interval 14-210 days prior to the inclusion (test) date, yes or no. We have added the following to methods, p. 6: "And finally, because vaccination and time from vaccination may affect our findings, we repeated the time-differentiated analyses by stratifying the logit model on vaccination status and time since vaccination (having received the latest dose (1st, 2nd or 3rd dose) in the time interval 14-210 days prior to the inclusion/test date, yes or no, i.e. similar categorization as in our recent study on vaccination and medical complaints¹⁹

		among the infected. Because of few observations and a likely low statistical power in such stratified analyses, these analyses were only performed for the outcome including any of the symptoms.” Please see results from these analyses at p. 16, and a brief discussion at p. 24.
Again in sensitivity analysis, you could match Delta and Omicron cases on at least age and sex (not just week) given apparent differences between these groups.	All our models are adjusted for age and sex and a range of other potential confounders, which would correct for any potential bias induced by the potential confounders. Further, in accordance with a suggestion by Reviewer 3, we have omitted the sensitivity analyses using conditional logit models. The conditional logit models were omitted because it provided limited additional information on top of the main analyses.	None performed.
“The mortality during follow-up was low (0.08% (95% CI=0.06-0.10), 0.03% (95% CI=0.01-0.06) and 0.04% (95% CI=0.00-0.06) for persons testing negative or positive with delta and omicron, respectively” – Is this correct? Although CIs are overlapping (just) it does seem that the test negative group may have higher mortality implying this group may be different to the others in ways that have not been captured (shielding for example). They also appear to have higher reporting of anxiety/depression than the Delta or Omicron groups. This should be discussed.	We agree.	We have revised parts of the discussion section, p. 23: “It is possible that our population consisted of particularly health-conscious persons who were highly prone to get tested and who were more prone to seek medical care after knowing they had been ill. Indeed, there were some important differences in baseline characteristics on seeking medical care (testing and health care use) and mortality that may impact on our findings through selection/collider stratification and/or confounder bias. We believe our methodological approach ensuring comparison of persons who were

		tested in the same calendar week, the inclusion of untested and untested + test negative in sensitivity analyses, as well as the adjustment for a range of covariates including health-seeking behaviour would limit these potential biases. Further, any differential mortality is unlikely to impact on our findings as it was below 0.2% for all study groups.“
Minor: I would include supplementary Fig 2 in the main part of the paper. This says 20-70 years, the text says 18-70 years.	We agree.	We have included S-Figure 2 in the main paper and it is now entitled Figure 1. 20-70 years was a typo and has been corrected to 18-70 years. The numbers of included individuals at each stage in Figure 1 have been slightly altered due to the detection of a minor coding error and due to updates in our inclusion criteria (please see response and action to comments by Reviewer 3 on inclusion and comparison groups).
There is a hint in the conditional logit model (Suppl Table 5) that fatigue reporting is higher for Delta than Omicron (different to the continuous Cox model) – this should be commented on.	We agree we might have missed the higher estimate for fatigue among Delta than among Omicron infected. However, based on comments by Reviewer 3 suggesting this analysis was not warranted, we have omitted it.	We have checked specifically the difference in fatigue between Omicron and Delta in our new logit models with plotted predicted probabilities and estimates of group difference for each post-covid time period (acute, sub-acute and chronic). These models were based on more outcome data: 1) more reimbursement forms and medical records have been received from the general practitioners and emergency wards in Norway since the initial extraction of data in the current study (we updated our data extraction on August 10th 2022), and 2) the logit models include all outcome data

		(aggregated to present once a week yes or no), not only the first record as was the case in the conditional logit model. Because the newly presented results are based on more data, we are confident that there are no essential group differences between Omicron and Delta regarding post-covid prevalence of fatigue. The hint as described by the reviewer may have been an incidental finding. Still, we do find some important group differences in the prevalence of any complaint after 90 days, which was consistent through all the sensitivity analyses. Please see the results and discussion section.
What is the size of the untested group in Suppl Table 9?	The size of the untested group in the initial version of the manuscript was N=1 373 092. We agree this should have been denoted in S-Table 9.	Due to other reviewer comments, S-Table 9 has been replaced with S-Figure 2 and S-Figure 3. We have included the size of the untested group in the figure legends (N=1 180 716). The number of untested is altered in the current revision compared to the initial version. This is due to updated selection criteria and methods as a response to a comment on selection by Reviewer 3.
In limitations, there should be mention of potential bias from preferential sequencing of suspected Omicron samples over Delta.	We agree.	We have added to the limitations section: “A final limitation may be preferential sequencing of suspected Omicron samples over Delta. If present, we believe it had limited impact on our findings, as S-Figure 1 shows that the inclusion period covered the period with the greatest

		overlap between the variants (50-50 around December 24th 2021).“
This is not the only study to look at post-Omicron persistent symptoms, and other studies suggest persistence beyond 90 days which will be a continuing burden on health services (“this burden will cease when 90 days have passed”) – introduction and discussion need updating accordingly.	We have once again searched for studies comparing Omicron and Delta, or Omicron to other control groups, and could not find any studies for valid comparison to the current study. However, we agree we could be more nuanced in our description of burden on health services.	Please see revisions to the section on interpretation and relevance, discussion section, p. 22: “Overall, our findings suggest that the included post-covid complaints exist to a similar extent after Omicron as after Delta, at least for the acute and sub-acute post-chronic phases. However, we found indications that Omicron might be milder than Delta at 90 days after testing positive and beyond, in studies of any complaint and in studies of musculoskeletal pain. No group differences after 90 days could be observed for the assumed main post-covid complaints (fatigue and respiratory complaints) as defined by the World Health Organization ¹² (“persistent complaints, typically fatigue and shortness of breath, with unknown cause still present at 3 months from the onset”). Our findings suggest that Omicron and Delta will lead to a similar burden of such WHO-defined post-covid fatigue and shortness of breath in the long run, yet that there may be fewer visits with any post-covid complaint and fewer visits with musculoskeletal pain in the Omicron-infected than in the Delta-infected.”
The authors might refer to Whitaker M et al Nat Commun 2022, Apr 12;13(1):1957 which identified two clusters of symptoms	We agree.	We have included this important reference in our discussion section, p. 21:

post-SARS-CoV-2 infection for earlier variants, one characterised by fatigue and one by shortness of breath.		“Like previous studies, for example based on self-reported data or hospital data^{1,20}, we found that the risk of fatigue and shortness of breath was elevated for the infected compared to the non-infected.”
“novel insights into disease etiology of post-omicron” – this is not really looking at etiology rather natural history post infection?	We agree.	We have done the following revision, p. 23: “Thus, our findings may have some important combined clinical and public health messages. First, we provide novel insights into the natural medical history after infection with Omicron vs Delta, demonstrating the need to further study the onset, duration and severity of post-covid complaints following the omicron variant, e.g. using patient-reported or clinical data.”
Abstract: “Omicron was related with a similar, and no increased risk of musculoskeletal pain, cough, heart palpitations, anxiety/depression when compared to delta and when compared to test negative” – confusing wording, re-word “Compared with either test negative or Delta, people testing positive for Omicron did not have increased...”	We agree.	We have rewritten the abstract to focus on the Omicron vs Delta comparison. We also include data from the newly added logit models. Our results in the abstract now reads: “Studying 1 323 145 persons aged 18-70 years living in Norway with and without SARS-CoV-2 infection in a prospective cohort study, we found that persons with Omicron had similar risk of a range of specific post-covid complaints (fatigue, cough, heart palpitations, shortness of breath and anxiety/depression) as persons with Delta, from 14 to up to 126 days after testing positive, both in the acute (14 to 29 days), sub-acute (30 to 89 days) and chronic post-covid (≥90 days) phases. However, at 90 days or more after testing

		positive, persons with Omicron had lower risk of having any complaint (43 (95% CI=14 to 72) fewer per 10 000), as well as lower risk of musculoskeletal pain (23 (95% CI=2-43) fewer per 10 000) than persons with Delta.”
“sanger” should be “Sanger”	We agree.	Corrected.
“covid” should be “COVID-19”	We agree.	Corrected.
“Moreover, all participants in our study had a PCR” – typo for participants.	We agree.	Corrected.
Omicron and Delta should have capitals throughout.	We agree.	Corrected.
Reviewer 2 comments	Our response	Action
This is an interesting and timely paper as there has been much made in policy terms that ‘Omicron is milder’ and therefore causes less Long Covid. The effect of Omicron compared to Delta in acute COVID is heavily confounded by vaccination and prior infection, and not always in a positive way as the recent paper by Boynton and Altman shows. Given that Long Covid is starting to become the lasting legacy of the pandemic with 100,000s of long term disabled in many countries, whether Omicron, and presumably its various sub strains is less likely to cause Long Covid is an important issue for policy (or at least it should be).	Thank you for the summary and encouraging comments.	
What is important here is that this is a controlled observational cohort comparing, in the same space of time, PCR negative, PCR+ve Delta and PCR+ve Omicron, with follow up up to 126 days using routine	We agree.	Please see specific responses and actions to the comments beneath.

primary care electronic health record data. However, there are some issues, and these relate to power and to potential ascertainment bias in the outcome measure (failure = one of a set of ICPC2 codes for recognised Long Covid symptoms being recorded by the GP). Lets unpick these in turn:		
- Power issues. We don't have a section in the paper on power, but its important as the population is being split in three, then into three time periods, then into 7 sets of sub-symptoms, then there are those without tests available, or without data available. Although the analysis is based on person-days, there are 242,264 person days in the Omicron group at 90-126 days (supplementary table 8), assuming a full 126 days for each, that is less than 2,000 individuals - or at 5% (being current accepted estimate of Long Covid risk at 6 months) that would be 100 people split up between the 7 symptom sets. Its easy to see that all these sub group analyses are really underpowered. A rule of thumb would be 50 individuals minimum per cell. I don't think this issue is adequately addressed in the paper and I think the authors need to do two things 1. Do some post-hoc power calculations (correcting also for the multiple comparisons) 2. Restrict the main findings to those cells that are adequately powered, regardless of the result significance and recognise that all others are 'exploratory only'.	We agree that having sufficient statistical power to draw conclusions is important in observational studies as the current study. We also understand the concern of potentially underpowered analyses in our initial version of the paper. For reasons regarding power as mentioned by the reviewer and for a better ability to shed light on the relapsing-remitting nature of Long COVID as requested by Reviewer 1, we think a different statistical model including more outcome data and finer time intervals is warranted. Still, we agree that some of our results, particularly those for brain fog which had the fewest observations should be interpreted with care. We would politely like to refrain from doing post-hoc power calculations because of warnings against such analyses in the literature. Power calculations are very important in planning an empirical study to be done in the future. However, our data collection and study have already been performed, and the meaning and utility of power becomes much less clear. All statistical tests are already performed, and they can no longer be easily interpreted as the probability	Due in part to this comment and comments made by the other reviewers, we have revised the paper and results section by, firstly, including additional and updated data, with more observations. Second, we have removed the part of the Cox regression analysis, which was split by time, and replaced it by a logistic regression analysis with weekly outcome data. We also discuss potential power issues in the limitation sections of this paper, p. 24: “A third limitation may be misclassified and potentially underreported and/or underpowered (outcome) data, as briefly described above. For example, we had few observations of brain fog, and estimates should be interpreted with care. To face these challenges, we added an outcome including any of the specific complaints, consistently showing in the main and sensitivity analysis (stratified by vaccination status) that there might be greater differences between Omicron and Delta than found in analyses of each of the specific complaints.”

	of a desired future event. Please see more detail in work done by Dziak et al., 2020: (The Interpretation of Statistical Power after the Data have been Gathered. Curr Psychol.) In observational studies like ours, it is both recommended and general practice to report and focus on confidence intervals to provide information on statistical power and precision (Vandenbroucke et al. PloS Medicine 2007; Schulz & Grimes, Lancet 2005).	
- Ascertainment bias. Although in the Norwegian payment system GPs need to put at least one ICPC2 code into a consultation, there is by no means comprehensive recording of ALL symptoms present when a patient consults. Only coded data are analysed, not free text, so we are almost certainly hugely underestimating the actual prevalence of symptoms in this group of patients. This explains a very large part of the gap between incidence rates reported here and in patient-reported surveys. In addition I looked at REF 12 in detail, which is used to justify the statement that the GP records are a reliable source of symptom data. The study is based in practices in a single region who are already active in a research network. Altogether codes were not present in 144 of a total of 398 (36 %) ‘simple contacts’ (with no issuance of a prescription). Generally ICPC2 is a decent way of	We agree we might have ascertainment bias as described by the reviewer. We also agree that an analysis of having any of the symptoms could be a valuable addition to our work. However, it is likely to anticipate that the potential bias as described by the reviewer would be equal in the different exposure groups (i.e., causing non-differential misclassification of the outcome).	First, as a response to the reviewer comment and also other reviewer comments, we have provided a general discussion regarding ascertainment bias at p. 21-22: “It is possible that persons with shortness of breath (or with other complaints) refrained from contacting the physician for a second time, or that the physician did not bother recoding the same complaint, potentially leading to misclassification of complaints towards the later study periods. However, unless the widespread talk of long-covid leads to behavioral responses only among the infected, we would expect such time-differential misclassification to affect all study groups to an equal extent, i.e., it would have limited impact on our findings. Misclassification bias is a common threat to validity in all register-based research and in exchange, such

capturing symptom codes in primary care, but in Long Covid the use of P20 ‘memory disturbance’ isn’t really a good map to brain fog - which is largely a mental tasking problem. There is also a risk of systematic underreporting of data from multimorbid patients, since the KUHR database only imports the first two diagnoses from the reimbursement cards. One way around these issues is to look at an analysis of ‘any Long Covid symptom’. This would still be subject to ascertainment bias but would be better powered.		research may provide a good overview of the health service burden posed by a disease.” Second, we have added the “Any post-covid complaint” as an outcome throughout the paper. These analyses gave some interesting new results, which we have briefly discussed. Please see the action to the previous comment.
Overall I have some issues with the interpretation that the authors put on the results in the abstract and discussion. Firstly we are told that “no increased risk of musculoskeletal pain, cough, heart palpitations, anxiety/depression when compared to delta and when compared to test negative” - this is not reliable on account of the power and ascertainment issues. I also don’t think the statement ‘The omicron variant will likely lead to a temporarily increased burden on healthcare services.’ is warranted. Given the power and ascertainment issues we just don’t know that Long Covid is ‘temporary’ - in fact all the other data - especially UK ONS surveys suggest that for about half of patients it is not. Professor Brendan Delaney	We agree the interpretation of results could have been better supported by the data in our initial version of our manuscript. As a response to previous comments and to other reviewer comments, we have taken several actions that may improve the precision of results. Most importantly, we have included more outcome data and applied a model where all mentions (one per week or more, yes or no) are included, with the differences for Omicron vs Delta being studied for each of the major post-covid phases (acute, sub-acute and chronic).	The results and conclusion parts of the abstract now reads: “Studying 1 323 145 persons aged 18-70 years living in Norway with and without SARS-CoV-2 infection in a prospective cohort study, we found that persons with Omicron had similar risk of a range of specific post-covid complaints (fatigue, cough, heart palpitations, shortness of breath and anxiety/depression) as persons with Delta, from 14 to up to 126 days after testing positive, both in the acute (14 to 29 days), sub-acute (30 to 89 days) and chronic post-covid (≥ 90 days) phases. However, at 90 days or more after testing positive, persons with Omicron had lower risk of having any complaint (43 (95% CI=14 to 72) fewer per 10 000), as well as lower risk of musculoskeletal pain (23 (95% CI=2-43) fewer per 10 000) than persons with Delta. Our findings

Imperial College London		suggest that the acute and sub-acute burden of post-covid complaints on health services is similar for Omicron and Delta. The chronic burden may be lower for Omicron vs Delta when considering the experience of any complaint and musculoskeletal pain, but not when considering the experience of fatigue, cough, heart palpitations, shortness of breath and anxiety/depression.” Please note that we provide no conclusion for brain fog due to few observations. Results for the outcome are specifically addressed in the discussion section.
Reviewer 3 comments	Our response	Action
Magnusson et al. use Norwegian health registry data to compare the incidence of a set of complaints that commonly occur following Covid infection between individuals who were infected with the Omicron variant, individuals who were infected with the Delta variant and individuals that were tested and found negative. The study is based on seemingly high-quality data, and there is value in knowing the incidence of “soft complaints” of the type studied here following Covid-19 infection with different variants, but I do have certain concerns.	Thank you for the summary and encouraging comments.	
== Design == In the main analysis, the authors condition on being tested. This conditioning has	We agree.	We have added to the Discussion section, p. 23:

advantages – it seems to lower the probability of exposure misclassification - but also has disadvantages – most of all that it exposes the study to collider stratification bias. This is further supported by the fact that the “tested negative” group had higher mortality during the study period than both infected groups (!), suggesting that testing is a result of both being infected and of things that result in a higher risk of death (=it is a collider). The authors must acknowledge this in the Discussion.		“Indeed, there were some important differences in baseline characteristics on seeking medical care (testing and health care use) and mortality that may impact on our findings through selection/collider stratification and/or confounder bias. We believe our methodological approach ensuring comparison of persons who were tested in the same calendar week, the inclusion of untested and untested + test negative in sensitivity analyses, as well as the adjustment for a range of covariates including health-seeking behaviour would limit these potential biases. Further, any differential mortality is unlikely to impact on our findings as it was below 0.2% for all study groups.”
In the same context, the authors wisely perform a sensitivity analysis in which a negative test is not required for the control group. I have two concerns regarding this (important) analysis: first, it would be preferable to ignore testing altogether in this group and include both tested and untested, instead of specifically excluding the tested. Second, even if the current decision to include only untested is kept, the criteria should under no circumstance condition on being tested after the random inclusion date, which would result in clear selection bias. So, in this case, the sentence “never tested for SARS-CoV-2” in the Methods section should only pertain to the pre-study period.	We agree with the reviewer that there may have been sources of bias in our initial analyses and that our selection procedures could have been better described. In accordance with our previous studies of patterns in health care services use, we selected the individuals with negative tests from the population of having only negative tests, and the individuals who were untested from the population that was never tested (Magnusson et al., BMJ 2021). Thus, our analyses were conditioned on the future in both comparison groups, which is not appropriate in a prospective cohort study like the current study, where we aim to shed light on the etiology of health care visits (i.e. not only patterns of healthcare use). Further, we agree that a comparison group consisting of persons	First, we have updated our selection criteria and methods, of both the main analyses and the sensitivity analyses in a way that we do not condition on what happens after an individual is included. Please see revisions to the methods section p. 4: “We excluded all persons with previous positive PCR tests (up until December 7th 2021, to avoid pre-existing post-covid complaints), persons with unscreened positive tests and all persons who had a hospital contact from -2 to +14 days from the test date⁸.”

	testing negative and persons who were untested combined can be regarded to be more representative to the source population.	“Participants were categorized into three study groups based on their test result and date of testing: 1) persons with Omicron, 2) persons with Delta, and 3) persons who were non-infected (tested negative during the study period and/or earlier but allowed to test positive after the test date).” ..and revisions at p. 6: “The untested persons were included from their randomly assigned test date and were never tested prior to this date (they were allowed to have positive tests after the inclusion date but not required to).” Second, in a sensitivity analysis, we have added a comparison group consisting of both tested and non-tested individuals as described above. These revisions led to small alterations in the numbers of included persons in each study group. The revised inclusion criteria, together with other reviewer comments, also had consequences for our methods, please see Statistical analyses section and responses and action to comments on methodology, particularly the censoring part. Our main conclusion from these revisions to our sample was unaffected.
This important study does not address a causal question (i.e., because the exposure is not manipulable, and because the causal contrast is ill-defined for persons who	We agree the covariates cannot be called confounders. However, we think the exposure is manipulable in a hypothetical randomized controlled trial, i.e. when participants are	We think the phrasing “potential confounders” is the most correct in our study and have reworded at places where only “confounders” were used.

would not be infected by one or both variants at a given exposure level), and the authors wisely avoid causal language. Despite this, the variables adjusted for in the analysis are called confounders, which is a causal term. While this mishap is common in the literature, the phrase could be omitted altogether (perhaps replaced with “covariates”) to avoid any such error.	randomized to either being infected with Omicron or being infected with Delta. Such an RCT is off course not feasible or ethical as it may place the participants at severe health risk. Thus, we need to perform a prospective cohort study based on already collected observational data, where infection with either Omicron or Delta is non-random. Although we cannot know whether and to what extent confounding is present in such a study, we do know that confounding is usually an important source of bias in studies with similar design. Without the prespecified (causal) assumption that certain covariates could possibly affect the exposure and the outcome, there would be no reason to adjust for the covariates. If we were to adjust for covariates which we do not assume impact on the exposure and outcome, we would have a study with more predictive aims, as described by Shmueli - To explain or to predict? Statistical Science, 2010. We did not aim to develop a prediction model for whom would develop the post-covid complaints in the future Still, we agree with the reviewer that one should generally be cautious using causal language even in observational studies where causal inference would be the final goal.	
Outcomes were identified using primary care data only. I am not sure this is reasonable. Would inclusion of specialist care and hospitalizations not improve the accuracy of the outcomes? Is this data available?	We agree that the inclusion of specialist care and hospitalization might improve the accuracy of outcomes. For example, a primary care record could be combined with a specialist care record, and we would be more certain the patient had the complaint. Specialist care/hospital data are available. However, as	None performed.

	we have previously reported that testing positive and having a mild disease course does not increase all-cause or cause-specific specialist/hospital health care use when compared to testing negative (Skyrud et al., PLOS One, 2021), we chose to not include these data here. Another reason to not include these data is the fact that we have few observations for some outcomes, and that we might face challenges with statistical power in analyses for the different post-covid periods, as pointed out by Reviewer 2 if we were to include more accuracy to our outcome data. Outcomes in primary care only is informative as it may shed light on the isolated burden of Omicron vs Delta on the primary care services only.	
I could not understand the design in the important analysis of “time differentiated risks”. If a person was coded with an outcome during an early period, is he allowed to recur in a later period? If the answer is yes, then wouldn’t misclassification be a serious issue (e.g., because the codes persist in the EMR, or alternatively because the physician wouldn’t bother recoding the same problem, or alternatively because the person would not bother approaching his physician again with a not-so-treatable complaint such as “fatigue”). If the answer is no, then “dilution of susceptibles” could fully explain the reduction in hazard rates over time.	We agree that the analyses of time differentiated risks using Cox regression analyses could be challenging to interpret. Persons with an outcome in an early period were indeed allowed to recur in a later period, but not in the same period. We think this could have been better described, or (even better), being differently analysed, e.g. using a model with finer stratified follow-up time and allowance for repeated outcomes (as also pointed out by Reviewer 1). We also agree that these issues should be included in the discussion section.	We have provided outcome data that might be better interpretable than the original analyses of time differentiated risks, as described in the methods section, statistical analyses, p. 5-6: ...” Thus, to assess whether the post-covid complaints were more or less common in certain periods after positive test (the acute phase (14 to 29 days), the sub-acute phase (30 to 89 days) and the assumed chronic post-covid condition phase (90 to 126 days) as recommended in previous studies,^{17,18}), we estimated the group-wise weekly proportions having the outcome in question (with 95% CI) and plotted the predicted probabilities from a logit model with standard errors clustered on person

		level, adjusted for the same covariates as described above. In these analyses, all medical records were included (i.e. not only the first one as for the Cox regression analyses - rather, all visits every week were included and dichotomized into having the outcome in question that week, yes or no). For each post-covid phase, we also calculated the group difference in prevalence for persons infected with Omicron vs persons infected with Delta, by subtracting the estimate for persons with Omicron from the estimate for persons with Delta.” In addition, in the discussion section, p. 21-22, we have incorporated the potential source of bias in our discussion of deviating results for the Cox model vs. the logit models. Because estimates of shortness of breath potentially deviated in the original vs added analyses, we use this outcome as an example: “The different findings in the different models may be due to the Cox model including only the first mention of medical record, whereas the logit models include all records (averaged to 1 per week). Thus, the Cox model would systematically pick the earliest record of shortness of breath, which we find from Figure 4, S-Figure 4 and Table 3 are clearly elevated the nearer they come to the test date. It is possible that persons
--	--	--

		with shortness of breath (or with other complaints) refrained from contacting the physician for a second time, or that the physician did not bother recoding the same complaint, potentially leading to misclassification of complaints towards the later study periods. However, unless the widespread talk of long-covid leads to behavioral responses only among the infected, we would expect such time-differential misclassification to affect all study groups to an equal extent, i.e., it would have limited impact on our findings. Misclassification bias is a common threat to validity in all register-based research and in exchange, such research may provide a good overview of the health service burden posed by a disease. “
I could not find mention of the types of vaccines used. Is this mostly Pfizer? Moderna? mRNA in general? A mix? Given the documented negative association between vaccination and long-Covid rates, this could be an important predictor?	We agree this information should be provided. After March 11th 2021, only mRNA vaccines were given in Norway.	We have added the following to p. 5: “We checked specifically for potential confounding by vaccination status (the number of mRNA COVID-19 vaccine doses: 0, 1, 2 or 3 or more).”
Given the relatively mild associations found (notwithstanding shortness of breath), the authors must acknowledge ascertainment bias as a possible explanation for the differences from the uninfected. Given the wide-spread talk of long-Covid, both the patients and their physicians would be more likely to complain about and document outcomes such as fatigue following a known infection.	We agree these issues should be discussed. However, due to space restrictions and due to the finding of deviant estimates for shortness of breath across two of our main analyses and not for fatigue, we chose to use shortness of breath as an example.	We have added to the discussion section, p. 21-22: “It is possible that persons with shortness of breath (or with other complaints) refrained from contacting the physician for a second time, or that the physician did not bother recoding the same complaint, potentially leading to misclassification of complaints towards

		the later study periods. However, unless the widespread talk of long-covid leads to behavioural responses only among the infected, we would expect such time-differential misclassification to affect all study groups to an equal extent, i.e., it would have limited impact on our findings. Misclassification bias is a common threat to validity in all register-based research and in exchange, such research may provide a good overview of the health service burden posed by a disease.”
A covariate that would be interesting to explore is “time from vaccination”, as we now know that certain facets of immunity wane rather quickly following vaccination. One could stipulate that Delta infections occurred closer to person’s vaccination, which resulted in less post-Covid. This would be interesting as a covariate and also as an interaction term with the infecting variant (the main exposure), as one could hypothesize that time-from-vaccination is more important for the Delta variant, against which the vaccination is more effective in general.	We agree vaccination is an interesting topic in our study. However, the study of vaccination is also complex, as it is not random whom get vaccinated, with what dose and at what point in time. Please see our recent study (Methi et al., 2022, https://www.medrxiv.org/content/10.1101/2022.07.08.22277413v1 for a thorough study of vaccination and post-covid complaints. The study was based on similar routinely collected register data as the current study, yet we did not include SARS-CoV-2 variant. However, for the newly added outcome “any post-covid complaint” in the current study, we think it would be possible to do an explorative analysis stratified by vaccine status.	We have added to the methods section, p. 6: “And finally, because vaccination and time from vaccination may greatly affect our findings, we repeated the time-differentiated analyses by stratifying the logit model on vaccination status and time since vaccination (having received the latest dose (1st, 2nd or 3rd dose) in the time interval 14-210 days prior to the inclusion/test date, yes or no, i.e. similar categorization as in our recent study on vaccination and medical complaints¹⁹ among the infected. Because of few observations and a likely low statistical power in such stratified analyses, these analyses were only performed for the outcome including any of the symptoms.”

		Further, we have added to the end of the results section, p. 16, with reference to the online supplementary file: “Sensitivity analyses of any complaint stratified on vaccination showed minor group differences up to 90 days after positive test (S-Figure 5, S-Table 5). However, after 90 days, persons with Omicron who were not vaccinated with their last dose (1, 2 or 3) at 14 to 210 days before their inclusion (test) date (N=3997 (29.9%)) would have 81 (33-129) per 10 000 fewer cases with any post-covid complaint compared with persons with Delta with similar no such vaccination (N=9607 (40.4%)). Among vaccinated persons (1, 2 or 3 at 14 to 210 days before their inclusion (test) date), persons with Omicron (N=9368 (70.1%)) would have 36 (1-70) per 10 000 fewer cases with any post-covid complaint at 90 days or more, compared with persons with delta (N=14 160 (59.6%)).” And finally, please see a brief discussion of findings, p. 24. “To face these challenges, we added an outcome including any of the specific complaints, consistently showing in the main and sensitivity analysis (stratified by vaccination status) that there might be greater differences between Omicron and Delta than found in analyses of each of
--	--	--

		the specific complaints. Interestingly, the largest group differences were seen for the chronic post-covid period, with absolute risk difference magnitudes -43 (-72 to -14) per 10 000 for the whole cohort and -81 (-129 to -33) per 10 000 for unvaccinated and -36 (-70 to -1) per 10 000 for vaccinated. We believe these findings suggesting Omicron is similar to Delta in the acute and sub-acute post-covid phase, but milder than Delta in the chronic post-covid phase warrant more investigation in future studies with longer follow-up periods. Further, the study of vaccination against COVID-19 and post-covid complaints is complex due to potential collider bias and healthy vaccinee bias.¹⁹ Our findings by strata of vaccination can only be regarded as explorative and should be confirmed using more suitable methods for causal inference from observational designs.”
Though I hesitate to offer yet more scientific questions, it would be interesting to try and ascertain the severity of the infection during the exposure period (the first 14 days following diagnosis), and address the known hypothesis that more severe infections = more post-Covid.	We agree this is an important research question. However, the only feasible way we could possibly define initial disease severity would be hospitalization prior to or after the test date. In previous studies comprising the earliest SARS-CoV-2 variants, we indeed reported more all-cause and cause-specific healthcare use following COVID-19 related hospitalization than following COVID-19 not requiring hospitalization (Skyrud et al., PLOS One, 2021). Considering the low proportion with 1 or more PCR tests being hospitalized in the current study (7%), we are concerned such	None performed.

	an analysis would have limited validity due to few outcome observations and low statistical power.	
Tables 4 and 5 and all similar tables in the supplementary would be more useful as forest plots (like figures 1 and 2) with side-by-side columns for each time period.	We agree.	The analyses forming the base for Tables 4 and 5 (time-differentiated risks calculated from Cox regression models) have been replaced with analyses of the weekly share visiting primary care with our outcomes. From logit models, we could estimate and plot the predicted probabilities. Thus, Tables 4 and 5 have been omitted and replaced with timeline plots (Figure 4 and S-Figure 4). The sensitivity analyses (different comparison groups and censoring in Cox regression) are now presented as forest plots rather than as tables in the online file (S-Figure 2, S-Figure 3).
== Analysis == The main purpose of the study, as suggested by the title and the first line of the abstract, is comparing outcomes following Delta vs. outcomes following Omicron. This would make the information in Figure 2 (contrasting Omicron vs. Delta) the main result of the paper. Despite this, when results are cited in the Abstract and Results section text, the results cited are from Figure 1 (contrasting Omicron and Delta vs. uninfected). The authors should explicitly define their main study question and report the main results accordingly.	We agree.	We have switched places for Figure 1 and 2 and have rewritten the abstract in a way that it focuses more on the Omicron vs Delta comparison.
In general, one should not report a difference of two parameters without	We agree.	We have rephrased, and now summarize with direct comparisons:

directly contrasting them. In this regard, statements such as “The risk of complaints was the highest in the acute phase (14 to 30 days) and decreased for both variants in the sub-acute (30 to 90 days) and assumed chronic post-covid condition (here: 90 to 126 days) phases” in the Discussion section do not seem well founded without a direct comparison being made and its uncertainty (i.e., confidence intervals) reported.		“In this population-based prospective cohort study, we found that persons with Omicron had similar risk of a range of specific post-covid complaints as persons with Delta, both in the acute (14 to 29 days), sub-acute (30 to 89 days) and chronic post-covid (≥ 90 days) phases. However, at 90 days or more after testing positive, persons with Omicron had lower risk of having any complaint (43 (95%CI=14 to 72) fewer per 10 000), as well as lower risk of musculoskeletal pain (23 (95% CI=2-43) fewer per 10 000) than persons with Delta. “
The authors at times commit the “absence of evidence is not evidence of absence” fallacy, for example when stating “no increased rates of ... and brain fog in any of the post-covid clinical phases” in the Discussion section, when the CI estimated was 0.68-1.86, which is a noisy estimate that is also compatible with a very strong effect (86% increased risk!). The authors should rephrase more cautiously.	We agree.	We have rephrased to a more cautious interpretation, and updated the summary based on our new analyses, please see our action to the previous comment. We have emphasized the low number of observations of brain fog in the discussion section. Because estimates were inconclusive, brain fog is now not included in the main conclusion or abstract conclusion. However, considering previous reports of brain fog being a post-covid complaint, we believe it is of interest to report estimates for this outcome (Blomberg et al., Nature Medicine, 2021). Few observations could be an interesting observation on its own.
The main analysis in the paper consists of Cox proportional hazards models. As	Thank you for providing this informative and concise reference. We agree with the reviewer.	First, we have added the following to the Methods section, p. 5:

executed in this study, these models assume proportional hazards, which are unlikely to (and some say, cannot possibly) be true in real data. The authors should heed the advice of Stensrud et al. (https://jamanetwork.com/journals/jama/article-abstract/2763185), accept that average HRs are being reported, and change the analysis accordingly to correct the standard errors. In this context, it should be noted that stratification by test week only “solves” possible non-proportionality that results from the test week. The text is not clear about that.		“The stratification ensured that there was no possible non-proportionality of hazards resulting from the test week (although there may be violation of the assumption of proportional hazards for other variables).” Second, we have added to the same page: “Still, the hazard ratio estimate from a Cox proportional hazards model should only be regarded as a weighted average of the time-varying hazard ratios, i.e. a summary of the treatment effect during the follow-up.¹⁶” Third, we have bootstrapped the confidence intervals from the Cox regression analyses. And finally, we have supplemented with reports of effect measures directly calculated from absolute risks, p. 5-6: “Thus, to assess whether the post-covid complaints were more or less common in certain periods after positive test (the acute phase (14 to 29 days), the sub-acute phase (30 to 89 days) and the assumed chronic post-covid condition phase (90 to 126 days) as recommended in previous studies,^{17,18}), we estimated the group-wise weekly proportions having the outcome in question (with 95% CI) and plotted the predicted probabilities from a logit model
--	--	---

		with standard errors clustered on person level, adjusted for the same covariates as described above. In these analyses, all medical records were included (i.e. not only the first one as for the Cox regression analyses - rather, all visits every week were included and dichotomized into having the outcome in question that week, yes or no). For each post-covid phase, we also calculated the group difference in prevalence for persons infected with Omicron vs persons infected with Delta, by subtracting the estimate for persons with Omicron from the estimate for persons with Delta.”
I could not find mention in the paper of what happens to persons in the “tested negative” group if they are found positive in a different test during the follow-up period. Their data should be censored, if that were not done. Regardless, it should be reported.	We agree that censoring at positive test could have been better described and handled in our study. We did not apply such censoring in our initial version of the study because the censoring would only be possible for specific individuals in the comparison group who are not representative for the total group of exposed, resulting in a sort of dependent censoring (Jackson et al., Stat Med, 2014). More specifically, knowing that infection with the Omicron variant comprised 80% of individuals on December 31st 2021 (S-Figure 1), rising even further into January 2022, we would also know that close to all individuals who tested positive after testing negative, tested positive with the Omicron variant, further strengthening the dependent censoring. Further, with the knowledge that 1) Omicron is known to result in a milder initial disease course than previous variants (Maslo et al.,	We have added a sensitivity analysis in which non-infected individuals with a later positive test were censored from their date of positive test and onwards. As expected, the effect estimates contrasting Omicron and Delta to non-infected with censoring were generally higher than in the analyses contrasting Omicron and Delta to non-infected without censoring. Please see Figure 3 and S-Figure 2 vs S-Figure 3 (Cox regression analyses), and Figure 4 vs S-Figure 4 (plotted proportions over time) for comparison. In addition, we have added a discussion of the two approaches, p. 23-24 in Discussion section: “A second limitation may be that the 10-18% who tested positive after being

	JAMA, 2022), probably resulting in less anxiety and less testing in the population, 2) mass vaccination with the 3rd dose mRNA vaccine occurred in Norway begin January 2022 (Norwegian Institute of Public Health), probably resulting in fewer tests, 3) test criteria got milder throughout the follow-up period (Norwegian Government's timeline of pandemic guidelines and restrictions), also resulting in fewer PCR tests but more home/antigen tests (we had no access to test results from home/antigen tests), we can infer that only the most severe Omicron cases with some specific characteristics would have PCR test in place of or in addition to an antigen test during the follow-up period. Thus, such censoring of observations for these (unrepresentative) individuals might be indicative that they are more likely to fail more quickly, which might inflate effect estimates in comparisons between Omicron and Delta and the non-infected. As there are no universally applicable methods for handling such issues without introducing more complexity to the interpretation of effect estimates (e.g. imputation (Jackson et al., Stat Med, 2014) or inverse probability weighting or similar (Willems et al., Stat Methods Med Res, 2018)), we believe that reporting the proportion having positive test for each exposure group, as well as conducting analyses with and without censoring could shed light on the impact on effect estimates (Shih, Trials, 2002).	included with a negative test or no test were unrepresentative to the source population, introducing differential loss to follow-up. More specifically, knowing that infection with the Omicron variant comprised 80% of individuals on December 31st 2021 (S-Figure 1), rising even further into January 2022, we would also know that close to all individuals who tested positive after testing negative, tested positive with the Omicron variant, further strengthening the dependent loss to follow-up. Further, with the knowledge that 1) Omicron is known to result in a milder initial disease course than previous variants,⁴ probably resulting in less anxiety and less testing in the population, 2) mass vaccination with the 3rd dose mRNA vaccine occurred in Norway begin January 2022,²¹ probably resulting in fewer tests, and, 3) test criteria got milder throughout the follow-up period,²² also resulting in fewer PCR tests but more home/antigen tests (we had no access to test results from home/antigen tests), we can infer that only the most severe Omicron cases with some specific characteristics would have PCR test in place of or in addition to an antigen test during the follow-up period. Censoring these individuals from their date of positive test might violate the assumption of independent censoring, as a participant could be lost to follow-up because one of the outcomes was about to occur. Because
--	---	--

		our main study aim was comparing Omicron and Delta, for which censoring at positive test was not an issue, we chose to present proportions becoming infected during follow-up, as well as conducting analyses with and without censoring of observations from the date of positive test and onwards.¹⁴ As expected, the estimates from analyses with censoring at positive test were higher than estimates from analyses without such censoring (Figure 3 vs S-Figure 3 and Figure 4 vs S-Figure 4). We believe the alternatives to handle dependent censoring, e.g. imputation²³ or inverse probability weighting²⁴ would add unnecessary complexity to our study without contributing to responding to our main research question.”
I could not understand the analysis that was performed with conditional logistic regression. I understood that some sort of matching was done, but between whom? And how was censoring handled in this context? Given that I consider stratification by calendar week as sufficient for the concern of confounding by calendar time, I am not sure this analysis is warranted.	We agree this analysis, with matching of cases (persons with the outcome in question) and controls (persons without the outcome in question, two controls per case) on their calendar week of testing might contribute with limited information above what is provided through the Cox regression analyses.	Considering the many good suggestions for alternative / sensitivity analyses in the current revision, we decided to omit the conditional logistic regression analyses to make space for the new analyses.
In the first paragraph of the Results section, the authors report the incidence proportion of mortality in the different groups with 95% confidence intervals “calculated based on Wilson”. I do not know who Wilson is, there is no citation attached to this sentence,	We agree.	We have added the following to the introduction of Statistical analyses section, Methods, p. 5: “First, we described the study groups on baseline and follow-up characteristics

and in general explanations of methodology belong in the statistical analysis section.		using means with standard deviations, numbers observed with proportions and proportions with 95% confidence intervals based on Wilson.¹³” We have added the reference to Wilson’s confidence intervals to the reference list: Wilson, E. B. 1927. Probable inference, the law of succession, and statistical inference. Journal of the American Statistical Association 22: 209–212. https://doi.org/10.2307/2276774.
In table 1, median [IQR] for age would be more helpful.	We agree.	Revised.

REVIEWER COMMENTS

Reviewer #1 (Remarks to the Author):

The authors have responded appropriately to my concerns. A few remaining points:

- the authors chose not to carry out analyses of effects on symptom reporting post vaccination - this should be included in Discussion as additional limitation or future work.

Minor:

Line 201 - "The results were confirmed in sensitivity analyses" "similar" rather than "confirmed"

Line 220 "Time-differentiated shares having.." Re-word - what does "shares" mean in this context?

Line 310 "natural medical history after infection with Omicron vs Delta" - not quite right as it is natural history following Omicron or Delta against background of population immunity from vaccination/prior infection

Line 343 -"test criteria got milder.." What does this mean?

Figs - Omicron, Delta not consistently with CAP O, D in Figure labelling

Reviewer #2 (Remarks to the Author):

Thank you for asking me to review the revised paper. This is a careful and responsive revision to meet the reviewers comments. On balance this is a useful paper. I still have some reservations about power and sample size, but I accept that these are now clearly noted in the discussion section. 'Somewhat of an answer' is better than no answer at all and the results are clear, the discussion through and the paper well written and balanced.

I have no additional comments.

Point-to-point response to the reviewers, NCOMMS-22-23326-T (R2): Post-covid medical complaints following infection with SARS-CoV-2 Omicron vs Delta variants.
Magnusson et al., 2022.

We would like to thank the expert reviewers for having performed a careful review and consideration of our study, which we think has greatly contributed to further improve the quality of our work. Please see the detailed point-to-point responses and actions to the reviewers' comments beneath. Our page/table/figure references refer to the revised version of the manuscript *with* marked changes.

Reviewer 1 comments	Our response and action
The authors have responded appropriately to my concerns. A few remaining points:	Thank you.
- the authors chose not to carry out analyses of effects on symptom reporting post vaccination - this should be included in Discussion as additional limitation or future work.	We have added to p. 7. "For example, future studies could look into the effects on symptom reporting post vaccination."
Minor: Line 201 - "The results were confirmed in sensitivity analyses" "similar" rather than "confirmed"	Corrected.
Line 220 "Time-differentiated shares having.." Re-word - what does "shares" mean in this context?	We agree this was unclear. We have reworded the heading into: "Proportions having post-covid complaints in the different post-covid periods"
Line 310 "natural medical history after infection with Omicron vs Delta" - not quite right as it is natural history following Omicron or Delta against background of population immunity from vaccination/prior infection	We have revised the sentence into (changes in italics): "...natural medical history after infection with Omicron vs Delta in a population where the majority is vaccinated. "
Line 343 -"test criteria got milder.." What does this mean?	We have rephrased into "test criteria became less strict throughout the follow-up period, ²² embracing fewer and hence resulting in fewer PCR tests but more home/antigen tests.."
Figs - Omicron, Delta not consistently with CAP O, D in Figure labelling	Corrected.
Reviewer 2 comments	Our response and action
Thank you for asking me to review the revised paper. This is a careful and responsive revision to meet the reviewers comments. On balance this is a useful paper. I still have some reservations about power and sample size, but I accept that these are now clearly noted in the discussion section. 'Somewhat of an answer' is better than no answer at all and the results are clear, the discussion through and the paper well written and balanced. I have no additional comments.	Thank you.